# PRC1 collaborates with SMCHD1 to fold the X-chromosome and spread Xist RNA between chromosome compartments

Chen-Yu Wang [1,2], David Colognori[1,2,3], Hongjae Sunwoo [1,2,3], Danni Wang [1,2] & Jeannie T. Lee [1,2]

X-chromosome inactivation triggers fusion of A/B compartments to inactive X (Xi)-specific structures known as S1 and S2 compartments. SMCHD1 then merges S1/S2s to form the Xi super-structure. Here, we ask how S1/S2 compartments form and reveal that Xist RNA drives their formation via recruitment of Polycomb repressive complex 1 (PRC1). Ablating *Smchd1* in post-XCI cells unveils S1/S2 structures. Loss of SMCHD1 leads to trapping Xist in the S1 compartment, impairing RNA spreading into S2. On the other hand, depleting Xist, PRC1, or HNRNPK precludes re-emergence of S1/S2 structures, and loss of S1/S2 compartments paradoxically strengthens the partition between Xi megadomains. Finally, Xi-reactivation in post-XCI cells can be enhanced by depleting both SMCHD1 and DNA methylation. We conclude that Xist, PRC1, and SMCHD1 collaborate in an obligatory, sequential manner to partition, fuse, and direct self-association of Xi compartments required for proper spreading of Xist RNA.

[1] Department of Molecular Biology, Massachusetts General Hospital, Boston, MA, USA. [2] Department of Genetics, The Blavatnik Institute, Harvard Medical School, Boston, MA, USA. [3]These authors contributed equally: David Colognori, Hongjae Sunwoo. Correspondence and requests for materials should be addressed to J.T.L. (email: lee@molbio.mgh.harvard.edu)

Mammalian chromosomes show a distinct topological organization. At one level, chromosomes are partitioned into A and B compartments in interphase nuclei, with gene-rich, active chromatin partitioned into the A compartment, and gene-poor, inactive chromatin into the B compartment[1]. At a finer level, chromosomes are segregated into "topologically associated domains" (TADs) on a megabase scale[2,3]. Genetic elements interact frequently within TADs, but are insulated from each other between TADs. How these chromosome structures are constructed is an area of intense investigation. Evidence suggests that TAD formation requires two architectural proteins—cohesin and CCCTC-binding factor (CTCF)[4,5], with cohesin progressively extruding chromatin loops until reaching a pair of convergent CTCF sites at defined TAD borders[6,7].

Much less is known about how chromosome compartments are constructed. As compartments persist in the absence of CTCF or cohesin[4,5,8,9], compartmentalization occurs via a mechanism distinct from that of TADs. An important feature of chromosome compartments is their correlation with chromatin states[1,6]. Thus, it is thought that compartments are driven by self-association between the same type of chromatin, possibly in a biophysical process known as "liquid–liquid phase separation" (LLPS)[10,11]. Indeed, many chromatin-associated factors phase separate[10–13]. However, evidence that these factors are required for forming large-scale chromosome compartments is currently lacking.

X-chromosome inactivation (XCI), the mechanism by which mammals balance dosage of X-linked genes between males and females, is a powerful model to study chromosome architecture[14–19]. During XCI, the inactive X chromosome (Xi) undergoes massive reconfiguration into a unique structure distinct from the active X chromosome (Xa). For instance, the Xi acquires two large interacting domains separated by a tandem repeat locus, Dxz4[6,20–22]. Concurrent with formation of these "megadomains" is global attenuation of TADs[20–22]. Central to this process is Xist, a long noncoding RNA that spreads along the Xi, recruiting silencing factors and inducing a global suppression of TADs[21–24].

Structural reorganization of the Xi also occurs at the level of compartments. The Xi is unique in being "compartment-less," without the A/B compartmentalization that characterizes all mammalian chromosomes[22]. By ablating an Xi-enriched protein, structural maintenance of chromosomes hinge domain containing 1 (SMCHD1)[25–27], in pre-XCI cells and observing effects during de novo XCI, we recently identified a hidden layer of Xi organization—S1/S2 compartments[28]. This Xi-specific structure marks a transition in the progression from A/B compartments to the "compartment-less" structure. We proposed a stepwise folding origami model, in which A/B compartments are conjoined into S1/S2 compartments, which are then merged by SMCHD1 to create the Xi super-structure. During de novo XCI, failure to merge compartments results in compromised spreading of Xist RNA, segmental erosion of heterochromatin, and failed silencing of ~43% of genes normally subject to XCI[28]. Thus, S1/S2 compartments provide an example where the dynamic control of compartment structure contributes to gene regulation. Although it is clear that the merging of S1/S2 compartments relies on SMCHD1, the mechanism underlying their formation is presently unknown.

Here we investigate the molecular requirements of the stepwise folding mechanism. We show that merged Xi compartments can be unfolded. Ablating Smchd1 in post-XCI cells causes re-emergence of transitional S1/S2 compartments. SMCHD1 deficiency also traps Xist RNA in the S1 compartment, leading to aberrant enrichment of H3K27me3 on the Xi. Furthermore, depleting Xist, heterogeneous nuclear ribonucleoprotein K (HNRNPK), or polycomb repressive complex 1 (PRC1) prevents S1/S2 compartmentalization. These results reveal a dependence on PRC1 for creation of S1/S2 structures and a role of SMCHD1 in facilitating the spreading of Xist RNA between S1/S2 chromosome compartments.

## Results

**Post-XCI Smchd1 ablation unveils S1/S2 compartments.** We began by asking if SMCHD1 is required once the Xi is already established. We disrupted Smchd1 in mouse embryonic fibroblasts (MEFs), a cell type in which XCI is completed (Supplementary Fig. 1a–c). To probe three-dimensional (3D) organization of the Smchd1$^{-/-}$ Xi, we performed allele-specific in situ Hi-C on wild-type (WT) and Smchd1$^{-/-}$ clones, taking advantage of >600,000 sequence polymorphisms between the Xa of Mus castaneus (cas) origin and the Xi of Mus musculus (mus) origin to call alleles[29]. As predicted, the Xa's of Smchd1$^{-/-}$ and WT clones showed no significant differences between one another (Supplementary Fig. 2a).

On the other hand, the Xi showed striking differences. Whereas the WT Xi exhibited two megadomains separated by the boundary element, Dxz4 (Fig. 1a, left)[6,17,20–22,30], the Smchd1$^{-/-}$ Xi developed a heterogeneous pattern within each megadomain (Fig. 1a, right). Its Pearson correlation map showed a distinct checkerboard pattern suggesting smaller compartment structures appearing under the megadomains (Fig. 1b, right). This pattern was distinct from the A/B compartmental pattern of the Xa (Supplementary Fig. 2a, bottom). Principal component analysis (PCA) verified the changes in Smchd1$^{-/-}$ cells. Indeed, principal component 1 (PC1) showed that the Smchd1$^{-/-}$ Xa retained the ~63 A/B compartments that characterize the WT Xa ($r = 0.94$) (Supplementary Fig. 2b), but the Smchd1$^{-/-}$ Xi lost the compartment-less character of the WT Xi. Instead of the two large compartments that characterize the WT Xi, the mutant Xi exhibited ~25 compartments averaging 4.1 Mb in size (Fig. 1c). Notably, the 25 compartments were highly similar to S1/S2 compartments that occur transiently during de novo XCI ($r = 0.84$, Fig. 1d, Supplementary Fig. 3a–c)[28]. Thus, SMCHD1 loss is sufficient to unmask underlying S1/S2 compartments, and the folding of the Xi compartments appears to be reversible.

Chromatin compartments generally align with epigenetic landscapes[1,6,31,32]. Accordingly, on the Xa, A compartments correlated with active H3K4me3 marks in both WT and Smchd1$^{-/-}$ cells ($r = 0.52$) (Supplementary Fig. 2b). On the Smchd1$^{-/-}$ Xi, the re-emergent S1/S2 compartments remained correlated with the H3K27me3 density ($r = 0.77$) and Xist RNA binding ($r = 0.70$) that normally typifies the Xi (Fig. 1c, Supplementary Fig. 3d). Consistent with its role in merging S1/S2s[28], analysis of SMCHD1 binding patterns in MEFs indicated localization across both S1 and S2 domains (Fig. 1c, Supplementary Fig. 3d). Together, these data show that Smchd1 ablation unveils the transitional S1/S2 conformation, where Xist-rich (S1) and Xist-poor (S2) chromatin spatially segregate. We conclude that SMCHD1 is actively required to maintain the merging of S1/S2 compartments.

Interestingly, although S1/S2 compartments reappeared following Smchd1 ablation, the Smchd1$^{-/-}$ Xi did not lose its overall bipartite structure. Indeed, whereas PC1 reflected S1/S2 compartments, principal component 2 (PC2) captured the two megadomains (Fig. 1c). These megadomains were also discernible on the contact heatmap of the Smchd1$^{-/-}$ Xi, even though the Xi adopted a checkerboard pattern (Fig. 1a, b). These observations suggest that SMCHD1 deficiency does not eradicate the overall bipartite structure and merely unmask an S1/S2 organization merged on the WT Xi. S1/S2 compartments and megadomains

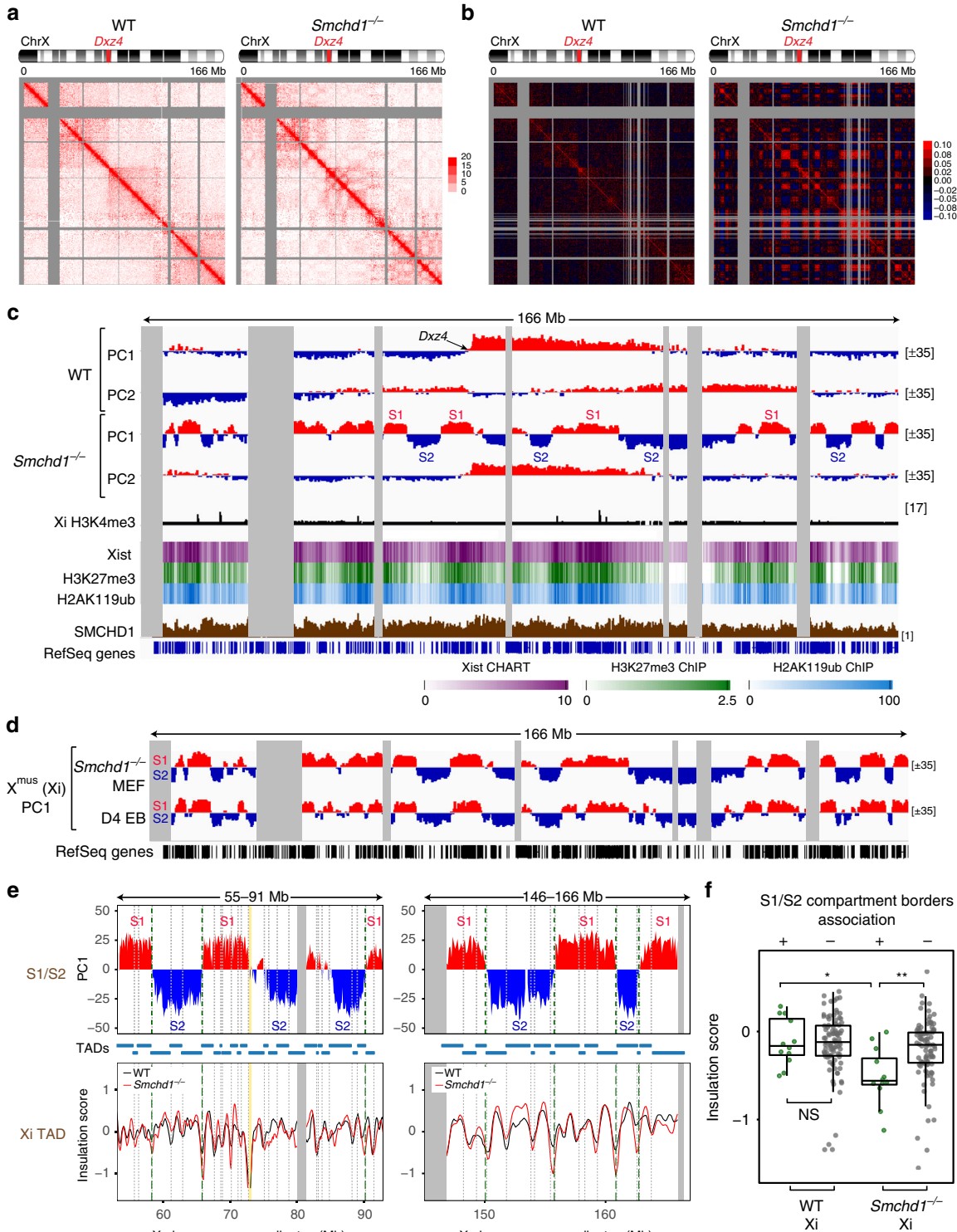

are therefore two different levels of 3D organization superimposed on the Xi.

**SMCHD1 is required to maintain Xi TAD attenuation.** Chromosome conformation studies have provided two different views of TAD organization on the Xi. Some studies indicated that TADs are dissolved during XCI[21,22]. However, the others using improved Hi-C methods have suggested that TADs remain on the Xi, albeit in an attenuated state[20,24,28,30]. One study from our lab furthermore showed that depleting SMCHD1 during de novo

XCI compromises TAD suppression[28]. Here we asked if ablating *Smchd1* in post-XCI cells causes a change to Xi TADs. Analysis of Hi-C contact maps at 100-kb resolution suggested that TADs also remained—in a weakened state—on the WT MEF Xi (Supplementary Fig. 4a). To quantify TAD strength, we computed insulation scores, a parameter that quantifies the frequency of chromatin interactions across a genomic region[22]. The insulation scores of the WT Xi fluctuated at smaller amplitude (Supplementary Fig. 4a, b), resulting in a steeper cumulative distribution curve than those of the WT Xa (Supplementary Fig. 4c, left). However, their local minima (representing TAD boundaries) and

**Fig. 1** Depleting structural maintenance of chromosomes hinge domain containing 1 (SMCHD1) leads to reappearance of S1/S2 compartments in post-X-chromosome inactivation (XCI) cells. **a** Depth-corrected chromatin interaction maps of the inactive X chromosome (Xi) in wild-type (WT) and $Smchd1^{-/-}$ mouse embryonic fibroblasts (MEFs) binned at 200-kb resolution. Gray-shaded areas, unmappable regions. Also see Supplementary Fig. 2a for the active X chromosome (Xa) maps. **b** The corresponding Pearson's correlation maps of the Xi in WT and $Smchd1^{-/-}$ MEFs binned at 200-kb resolution. Also see Supplementary Fig. 2a for the Xa maps. **c** Principal component 1 (PC1) and PC2 values of the Xi in WT and $Smchd1^{-/-}$ MEFs. Regions with positive PC1 values on the $Smchd1^{-/-}$ Xi represent the S1 (Xist-rich) compartment (red). Also shown are Xi-specific H3K4me3 chromatin immunoprecipitation followed by deep sequencing (ChIP-seq) peaks ($X^{mus}$, GSE33823), Xist CHART-seq (capture hybridization analysis of RNA targets with deep sequencing) (GSE48649), H3K27me3 ChIP-seq (GSE33823), H2AK119ub ChIP-seq (GSE107217), and SMCHD1 DNA adenine methyltransferase identification by sequencing (DamID-seq) (GSE99991) profiles in WT MEFs. Xist, H3K27me3, and H2AK119ub profiles were displayed as heatmaps, with scale bars shown below the tracks. Also see Supplementary Fig. 2b for the PCs of the Xa. **d** PC1 values of the Xi in $Smchd1^{-/-}$ MEFs and embryoid bodies formed after 4 days of differentiation of female WT mouse embryonic stem cells (D4 EB, early XCI) (GSE99991). Regions with positive PC1 values represent the S1 (Xist-rich) compartment (red). **e** Comparison of the compartment profile (PC1) (top) of the $Smchd1^{-/-}$ Xi and the topologically associated domain (TAD) insulation profiles (bottom) of the WT (black) and $Smchd1^{-/-}$ Xi (red) at two representative X-linked regions. TADs (as defined in Dixon et al.[2]) were depicted as blue bars between plots and as dashed lines (TAD boundaries) within each plot. Green dashed lines, the borders of S1/S2 compartments. Yellow-shaded area: $Dxz4$. Gray-shaded areas, unmappable regions. **f** Box plots comparing the insulation scores of the TAD boundaries associated with borders of S1/S2 compartments (green) and the other TAD boundaries (gray) on the Xi. Note that lower insulation scores indicate stronger insulation effects. P values are given by the Wilcoxon's rank-sum test. NS, not significant ($P > 0.05$). $*P = 0.0018$; $**P = 0.0015$. Midline, median. Top and bottom of the box, first and third quartile. Whiskers, extension from the top or bottom to the furthest datum within 1.5 times the interquartile range

maxima often coincided (Supplementary Fig. 4a, b), consistent with the idea that Xi TADs are attenuated.

When $Smchd1$ was ablated, however, there was an overall strengthening of Xi TADs (Supplementary Fig. 4a), indicating that SMCHD1 loss in the post-XCI stage also partially restores TADs. Insulation analysis revealed a greater fluctuation of the $Smchd1^{-/-}$ Xi (Fig. 1e, Supplementary Fig. 4a, b). In support, the insulation scores of the $Smchd1^{-/-}$ Xi exhibited a shallower cumulative distribution than those of the WT Xi (Supplementary Fig. 4d, right), consistent with stronger TADs. There was also an Xi-specific increase of TAD boundary strength (Supplementary Fig. 4e) and greater correlation of Xi TAD organization to the Xa (Supplementary Fig. 4f) in $Smchd1^{-/-}$ MEFs. We conclude that SMCHD1 is required to suppress TADs during both the establishment and maintenance phases of XCI.

TAD boundaries are known to demarcate functional chromatin domains[33,34]. Interestingly, most (92%) S1/S2 borders coincided with TAD boundaries (Fig. 1e, Supplementary Fig. 5). Moreover, the TAD boundaries that demarcate S1/S2 compartments tended to have stronger insulation effects (Fig. 1f). These findings indicate that TAD boundaries may play a role in demarcating the Xist-rich and Xist-poor compartments.

**SMCHD1 and DNA methylation synergize in maintaining XCI.** Because SMCHD1 deficiency during de novo XCI significantly affects Xi silencing[25,28,35,36], we asked if depleting SMCHD1 would reactivate the Xi in post-XCI cells. We performed RNA-sequencing (RNA-seq) on two independent $Smchd1^{-/-}$ and two WT MEF clones. Interestingly, expression of X-linked genes was mostly unchanged (Fig. 2a, Supplementary Fig. 6a, b). To examine the Xi specifically, we conducted allele-specific analysis on 216 genes subject to XCI in WT MEFs (216 genes passed our pipeline requirements) (Fig. 2b, Supplementary Data 1). Examining the fraction of Xi expression (%mus) and cumulative distribution plots revealed no significant difference between WT and $Smchd1^{-/-}$ MEFs (Fig. 2c). Thus, Xi silencing was untouched by post-XCI SMCHD1 depletion. We conclude that losing SMCHD1 alone is not sufficient to reactivate Xi genes.

It is known that multiple repressive mechanisms collaborate to maintain Xi silencing[37] and that reactivation requires perturbing multiple synergistic mechanisms[21]. Because DNA methylation has been identified as a major epigenetic mark to lock in the silent state[21,37,38], we treated $Smchd1^{-/-}$ cells with 5-aza-2′-deoxycytidine (Aza), a DNA-demethylating agent. Intriguingly, whereas Aza treatment caused only a mild Xi reactivation in WT MEFs

(Fig. 2d, left), it led to considerable Xi upregulation in $Smchd1^{-/-}$ MEFs, as demonstrated by a rightward shift of the %mus cumulative distribution curve for $Smchd1^{-/-}$ relative to WT cells, and also for dimethyl sulfoxide (DMSO) vs. Aza-treated $Smchd1^{-/-}$ cells (Fig. 2d, right panels). Heatmap analyses (Fig. 2e) and transcriptomic profiles (Fig. 2f) verified the Xi upregulation. By contrast, Aza treatment did not affect escapees, a subset of X-linked genes that resist XCI (Supplementary Fig. 6c [39,40], Supplementary Data 1). These data argue that SMCHD1 and DNA methylation operate synergistically to maintain Xi silencing.

As only a fraction of Xi genes was destabilized, we investigated factors correlating with the differential response of Xi genes. Genes in S1 and S2 compartments responded similarly (Supplementary Fig. 6d). It is known that SMCHD1 is uniquely required for de novo silencing of a specific subset of X-linked genes—the "Class I" genes—but is dispensable for inactivating other X-linked genes, including the Class II and Class III genes (defined in Methods and listed in Supplementary Data 2)[28]. Intriguingly, Class I genes were generally more prone to reactivation (Fig. 2g, Supplementary Fig. 7), although not all Class I genes were reactivated and a small fraction of Class II and III genes was also destabilized. Thus, the class of genes that were more sensitive to SMCHD1 loss during de novo XCI[28] is the same class that are more prone to reactivation in SMCHD1-deficient, post-XCI cells.

**Xist RNA is trapped in the S1 compartment in $Smchd1^{-/-}$ cells.** We previously reported segmental erosion of H3K27me3 domains on the Xi established without SMCHD1, most prominently in regions harboring derepressed Class I genes[28]. Here we asked if H3K27me3 also became eroded in post-XCI cells depleted of SMCHD1. At the cytological level, $Smchd1^{-/-}$ MEFs exhibited H3K27me3 foci co-localizing with Xist clouds (Fig. 3a, two Xist clouds seen because of tetraploidy), consistent with previous reports[25,27], giving the appearance that H3K27me3 on the Xi was not grossly affected. To determine if there might be regional erosion, we performed chromatin immunoprecipitation followed by deep sequencing (ChIP-seq) for H3K27me3. Gene bodies of escapees exhibited depletion of H3K27me3 in both WT and $Smchd1^{-/-}$ MEFs, as expected (Fig. 3b, $Kdm5c$, green-shaded area). On the other hand, Class I genes remained covered by this repressive mark, consistent with persistent gene silencing (Fig. 3b–d). Intergenic regions flanked by Class I genes, which also require SMCHD1 for de novo H3K27me3[28], were also

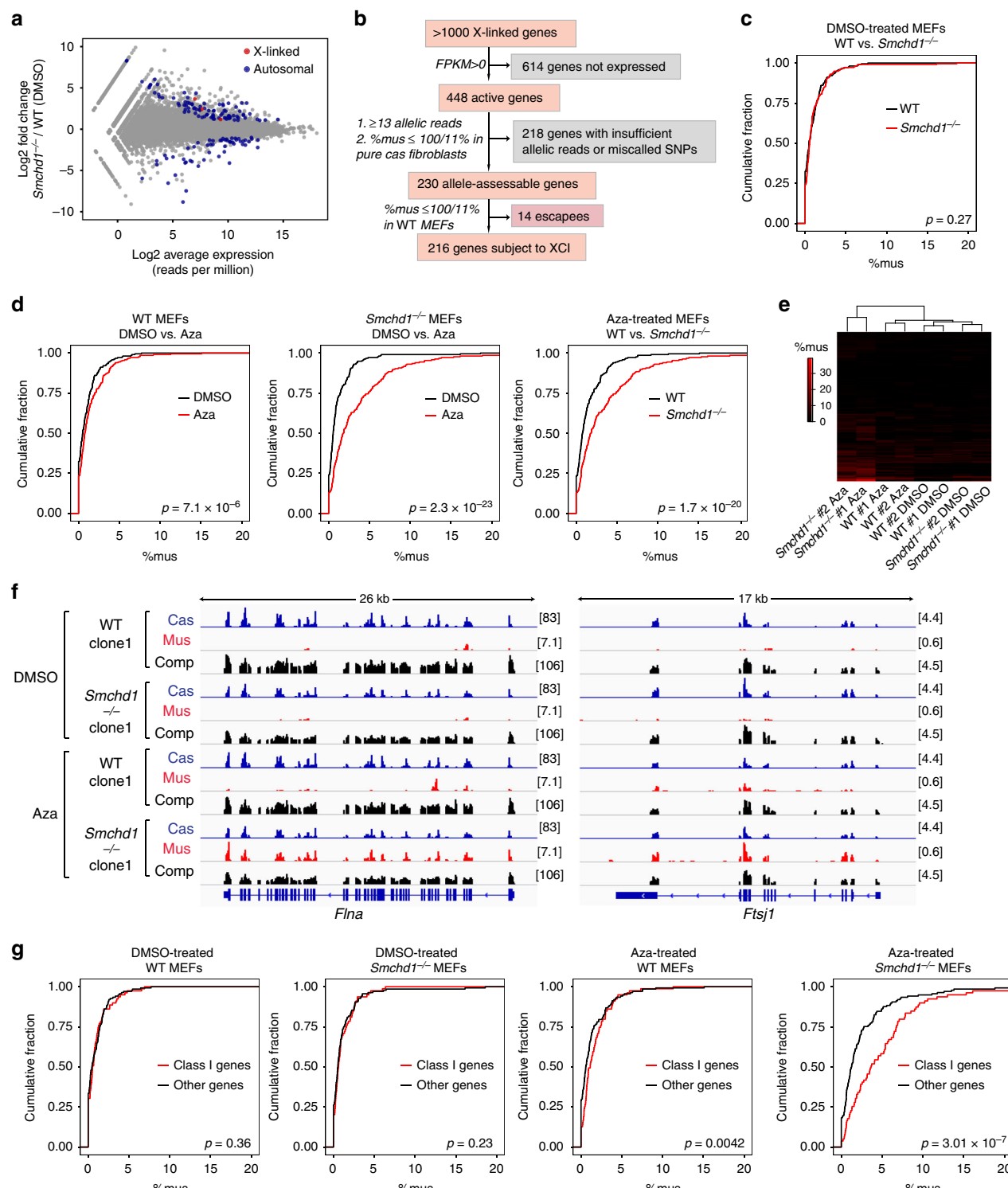

H3K27me3 enriched (Fig. 3b, c, e). Thus, SMCHD1 is not required to maintain H3K27me3 in Class I regions. Other classes of X-linked regions also do not require SMCHD1 to maintain H3K27me3 (Fig. 3d, e; black dots).

Not only was SMCHD1 not required, but its depletion actually resulted in an overall increase of H3K27me3 enrichment on the Xi relative to the WT Xi (Fig. 3b–e). This enrichment was Xi specific (Fig. 3f) and reproducible in two biological replicates (Supplementary Fig. 8a, b), and two independent normalization methods yielded the same conclusion (Supplementary Fig. 8c).

The finding was somewhat unexpected, as SMCHD1 generally promotes heterochromatin[28,35,36], although one previous study suggested a similar increase of H3K27me3[41]. There was no copy number difference between WT and Smchd1[−/−] Xi, arguing against X-chromosome aneuploidy as a cause (Supplementary Fig. 9). To determine if aberrant H3K27me3 accumulation is linked to the altered Xi structure, we investigated its correlation with S1/S2 compartments. As expected, H3K27me3 profiles correlated with S1/S2s (r = 0.92). Intriguingly, H3K27me3 was not homogeneously elevated on the Smchd1[−/−] Xi. Instead, this

**Fig. 2** Structural maintenance of chromosomes hinge domain containing 1 (SMCHD1) and DNA methylation synergistically maintain inactive X chromosome (Xi) silencing. **a** An RNA-sequencing (RNA-seq) MA plot comparing the gene expression profiles of wild-type (WT) and Smchd1−/− mouse embryonic fibroblasts (MEFs) treated with dimethyl sulfoxide (DMSO). Gray dots, genes not differentially expressed. Blue dots, autosomal differentially expressed genes. Red dots, X-linked differentially expressed genes. **b** Workflow showing the identification of genes subject to X-chromosome inactivation (XCI). Please see Supplementary Data 1 for the full list of genes subject to XCI and escapees. **c** Cumulative distribution plots (CDPs) of %mus for genes subject to XCI in DMSO-treated WT and Smchd1−/− MEFs. P values are given by the Wilcoxon's rank-sum test (paired, one-sided). **d** Cumulative distribution plots (CDPs) of %mus for genes subject to XCI in DMSO/5-aza-2′-deoxycytidine (Aza)-treated WT and Smchd1−/− MEFs. P values are given by the Wilcoxon's rank-sum test (paired, one-sided). **e** A heatmap showing the %mus of genes subject to XCI in eight different RNA-seq datasets, with unsupervised hierarchical clustering accurately grouping clones with the same genotype and treatment together. **f** Allele-specific RNA-seq coverage tracks of Flna and Ftsj1, two representative genes subject to XCI. cas, cas-specific reads (active X chromosome; Xa); mus, mus-specific reads (Xi). comp, all reads. To visualize rare mus reads originating from the Xi, the scales of mus tracks were set differently from the cas tracks. For simplicity, only the minus strand was shown. **g** CDPs comparing %mus of Class I genes vs. other genes subject to XCI. P values are given by the Wilcoxon's rank-sum test (unpaired, one-sided). Please see Supplementary Data 2 for the full list of Class I genes defined previously[28]

histone mark preferentially accumulated in the S1 compartment (Fig. 3g, h, Supplementary Fig. 8b), with the change in H3K27me3 (ΔH3K27me3) highly correlating with S1/S2s ($r = 0.9$).

Because Xist recruits PRC2 to the Xi[42], we asked if changes in Xist spreading underlie elevated H3K27me3. Because ablating Smchd1 does not affect Xist expression in post-XCI cells (Fig. 4a), increased H3K27me3 on the Smchd1−/− Xi was not caused by Xist upregulation. Although the Xist cloud morphology appeared cytologically normal in Smchd1−/− cells (Fig. 3a), molecular differences might be visible by CHART-seq analysis (capture hybridization analysis of RNA targets with deep sequencing)[43], an epigenomic method to map Xist binding to chromatin at high resolution. Indeed, we previously showed a local defect of Xist spreading into Class I regions during de novo XCI[28]. By contrast, when SMCHD1 was depleted post-XCI, Xist binding to Class I regions was preserved (Fig. 4b–d), consistent with H3K27me3 remaining enriched at these regions (Fig. 3b–d). Intergenic regions flanked by two Class I genes also remained bound by Xist (Fig. 4b, c, e). There is, however, a striking overall redistribution of Xist along the Smchd1−/− Xi (Fig. 4f). While the Xi remained broadly enriched with Xist, we observed heightened peaks and deepened valleys on the Smchd1−/− Xi (Fig. 4f). This may not reflect experimental variation of CHART, as difference in Xist distribution appeared in both biological replicates, and CHART in WT and Smchd1−/− MEFs seemed to have similar efficiency (Supplementary Figs. 8b, 10). To quantify changes in Xist binding, we subtracted WT Xist coverage from that of Smchd1−/− (ΔXist). Strikingly, the ΔXist profile correlated with ΔH3K27me3 ($r = 0.88$), suggesting that changes in Xist RNA localization likely underlie aberrant deposition of H3K27me3. As expected, Xist density on the Smchd1−/− Xi correlated with S1/S2 compartments ($r = 0.86$).

However, we noticed that the increase in Xist density on the Smchd1−/− Xi occurred specifically in the S1 compartment, whereas there was an obvious depletion in S2 (Fig. 4f, g), resulting in a ΔXist profile that correlated highly with alternating S1/S2 structures ($r = 0.8$). This was intriguing, as the Xist locus resides in the S1 compartment[28]. As such, it is tempting to speculate that the spatial segregation between S1 and S2 compartments on the Smchd1−/− Xi may compromise Xist RNA spreading from S1 to S2, resulting in a trapping of Xist in S1. This aberrant accumulation would explain the heightened H3K27me3 enrichment in S1. Thus, Smchd1 ablation affects Xist spreading during both de novo XCI[28] and the maintenance phase, but the effect is manifested differently during the two phases. In the former, Xist binding to Class I regions is compromised. In the latter, Xist can continue to spread across the S1 compartment (including Class I regions), but spreading into S2 is impaired.

**Xist is required to form S1/S2 compartments.** Next, we turned attention to understanding how S1/S2 compartments form. Because S1/S2s follow Xist-rich/-poor domains (Fig. 4f)[28], we investigated whether Xist is required. When Xist was deleted from the Xi after establishment of XCI in fibroblasts (XiΔXist)[44], we observed failed SMCHD1 targeting to the Xi, consistent with Xist being required to recruit SMCHD1 (Fig. 5a, b, Supplementary Fig. 11)[27,28]. We then performed in situ Hi-C on XiΔXist fibroblasts. Deleting Xist on the Xi had no apparent effects on the Xa (Supplementary Fig. 2a), as expected. However, XiΔXist had a striking absence of S1/S2 compartments (Fig. 5c, top-right), instead showing homogeneous long-range interactions within each megadomain. The pronounced checkerboard pattern of the Smchd1−/− Xi (Fig. 5c, bottom-middle) did not appear on the Pearson's correlation map of XiΔXist (Fig. 5c, bottom-right). Moreover, PC1 captured only two megadomains for XiΔXist (Fig. 5d, bottom tracks), rather than the ~25 S1/S2s observed for the Smchd1−/− Xi (Fig. 5d, middle tracks). These data indicate that, despite a failure of SMCHD1 recruitment, depleting Xist RNA does not unveil S1/S2 compartments. Xist RNA is therefore required to form S1/S2 compartments on the Xi.

**PRC1 partitions the Xi into S1/S2 compartments.** The Xi-specific S1/S2 compartments provided an excellent opportunity to explore the idea that compartments arise from co-segregation of chromatin with similar epigenetic states—possibly through a biophysical process known as "LLPS"[10,11,45]. We postulated that Xist RNA drives Xi compartmentalization by concentrating chromatin factors with self-associating properties. Moreover, such factors must be preferentially bound to Xist-rich S1 domains. One such candidate is PRC1. PRC1 catalyzes H2AK119 ubiquitylation (H2AK119ub), compacts chromatin, and is known to be enriched on the Xi in an Xist-dependent manner (Fig. 5f, Supplementary Fig. 11)[46–51]. Furthermore, PRC1 has the potential to polymerize[52] and promote small self-interacting domains at its target loci on autosomes[46,51,53,54].

Given these observations, we hypothesized that PRC1 action may underlie S1/S2 segregation. Indeed, in Smchd1−/− cells, H2AK119ub remained enriched on the Xi (Fig. 5g), indicating that PRC1 recruitment can still occur without SMCHD1. Allele-specific ChIP-seq revealed that H2AK119ub patterns mirrored the crest-and-trough pattern of Xist binding (WT, $r = 0.84$; Smchd1−/−, $r = 0.95$), correlating strongly with alternating S1/S2 profiles ($r = 0.86$)(Fig. 6a). As was the case for Xist and H3K27me3 densities, however, H2AK119ub density was heightened in the S1 compartment (Fig. 6a, b)—again consistent with the idea that Xist and its ensemble of recruited proteins being trapped in S1. Thus, Xist preferentially enriches PRC1 in the S1 domains in both WT and Smchd1−/− backgrounds.

We next asked if depleting PRC1 affects S1/S2 compartments in Smchd1−/− MEFs, where S1/S2s reemerged (Fig. 1). We

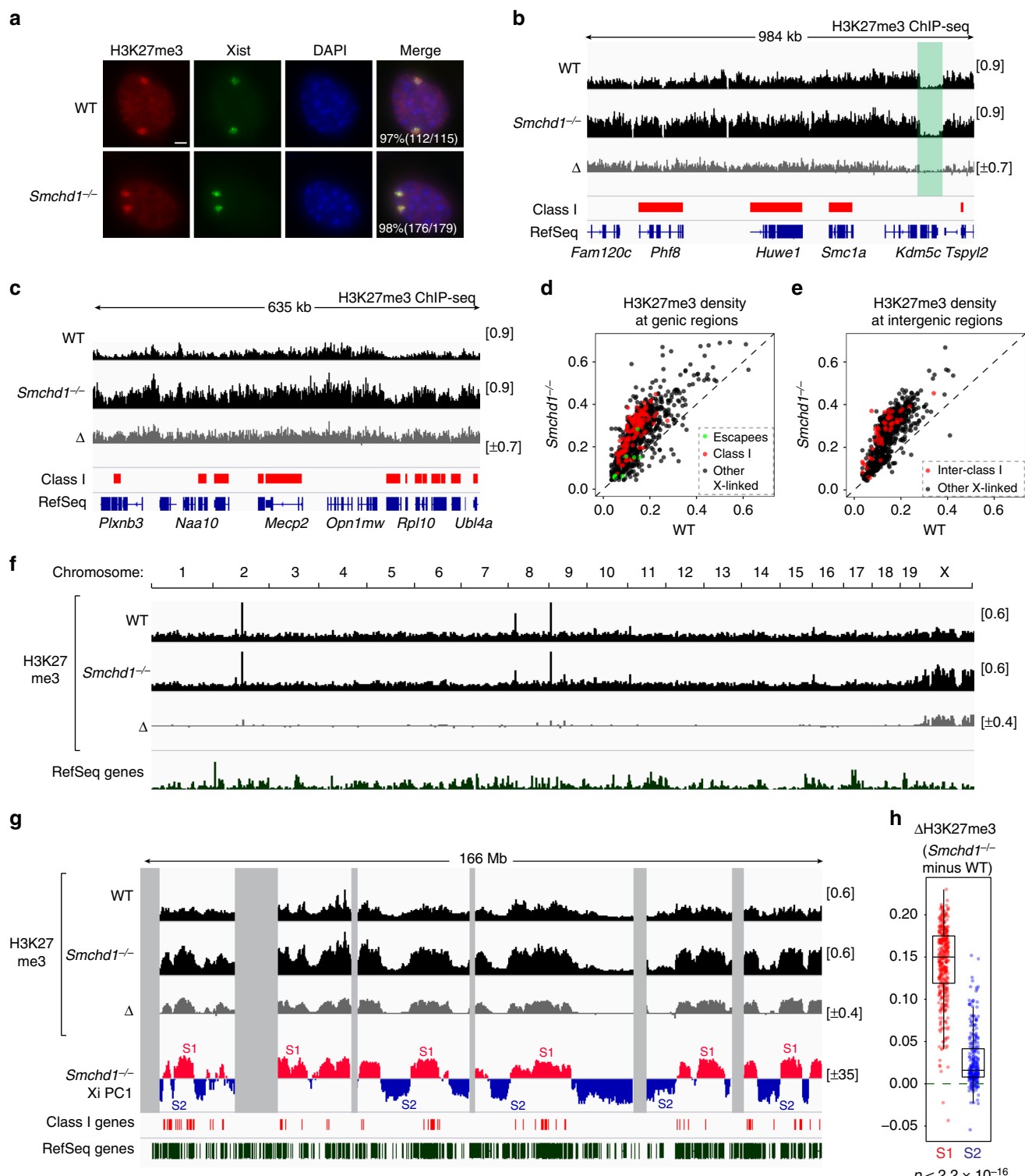

knocked down the catalytic subunits of PRC1—RING1A and RING1B ("PRC1 depletion")—and observed significantly attenuated H2AK119ub signals by western blot and immunofluorescence (Supplementary Fig. 12). In situ Hi-C revealed that, whereas the Xa showed no obvious changes (Supplementary Fig. 13), significant blunting of S1/S2 compartments was observed on the PRC1-depleted Xi (Fig. 6c). Contact heatmaps revealed that long-range interactions within each megadomain became more homogeneous (Fig. 6c, top), with the checkerboard pattern

on the Pearson's correlation map markedly weakened (Fig. 6c, bottom). In support, PC1 captured two megadomains on the Xi, but not compromised S1/S2s (Fig. 6d). Together, we conclude that PRC1 partitions the Xi into S1/S2 compartments.

To strengthen our conclusion, we also knocked down HNRNPK, a protein required for recruiting PRC1 to the Xi[24,55,56]. HNRNPK depletion recapitulates PRC1 depletion, causing a weakened checkerboard pattern on the *Smchd1*[−/−] Xi (Fig. 6c, right). Consistently, PC1 captured megadomains instead

**Fig. 3** Aberrant accumulation of H3K27me3 in the S1 compartment on the *Smchd1*[−/−] inactive X chromosome (Xi). **a** Immuno-RNA-fluorescent in situ hybridization (immuno-RNA-FISH) for H3K27me3 and Xist on wild-type (WT) and *Smchd1*[−/−] mouse embryonic fibroblasts (MEFs). Number of cells with H3K27me3 foci co-localizing with Xist clouds is shown. Scale bar, 5 μm. **b** H3K27me3 profiles for a representative X-linked region harboring several Class I genes and an escapee (*Kdm5c*). Red bars, Class I genes. Scales shown in brackets. Δ, *Smchd1*[−/−] minus WT. Note that we displayed the "comp" tracks (compiled from all reads) of H3K27me3 ChIP-seq (chromatin immunoprecipitation followed by deep sequencing), as most of the H3K27me3 signals are from the Xi. Allele-specific tracks have been deposited to Gene Expression Omnibus (GEO) (GSE116413). **c** H3K27me3 profiles for another representative X-linked region. **d** H3K27me3 density in gene bodies between WT (*x* axis) vs. *Smchd1*[−/−] (*y* axis) MEFs. Three categories of X-linked genes are shown. **e** H3K27me3 density in X-linked intergenic regions between WT (*x* axis) vs. *Smchd1*[−/−] (*y* axis) MEFs. Inter-Class I, intergenic regions flanked by two Class I genes. **f** H3K27me3 enrichment profiles across the entire genome. **g** H3K27me3 enrichment profiles across the X chromosome. Gray areas, unmappable regions. Also shown are the locations of Class I genes (red bars), and S1/S2 compartments in *Smchd1*[−/−] MEFs. **h** Box plots comparing the difference in H3K27me3 density between WT and *Smchd1*[−/−] cells of each 200-kb bin in S1 vs. S2 compartments. *P* values are given by the Wilcoxon's rank-sum test (unpaired, one-sided). Midline, median. Top and bottom of the box, first and third quartile. Whiskers, extension from the top or bottom to the furthest datum within 1.5 times the interquartile range

of attenuated S1/S2 compartments (Fig. 6d). This result supports a role for PRC1 in driving S1/S2 formation. Notably, a fainter checkerboard pattern remained visible on the HNRNPK-depleted Xi, and to a lesser degree, on the PRC1-depleted Xi (Fig. 6c, bottom). This residual S1/S2-like structure was too weak to be captured by PC1. However, PC2 exhibited an alternating profile reminiscent of S1/S2s (Fig. 6d). Together, these data suggest that depleting HNRNPK or PRC1 significantly compromised, but did not completely eradicate, the S1/S2 structure, likely because of incomplete knockdown (KD) of these factors.

**PRC1 partially obscures megadomains**. Interestingly, despite lacking two key structural factors, SMCHD1 and PRC1, insulation between the two Xi-specific megadomains became paradoxically strengthened. Notably, megadomain contours became sharper (compare Fig. 6c, WT vs. PRC1/HNRNPK KD), coinciding with reduced inter-megadomain contacts in PRC1/HNRNPK-depleted cells (Fig. 6e). By contrast, ablating *Smchd1* alone lacked this phenotype, suggesting an effect unique to depleting PRC1. Given similarly strengthened megadomains when either Xist or PRC1 was depleted (Fig. 5c–e), we suggest that Xist facilitates long-range interactions across the *Dxz4* boundary through self-association between PRC1-enriched chromatin, which thereby partially obscures megadomains. Loss of either Xist, HNRNPK, or PRC1 would therefore result in sharper megadomains. We conclude that PRC1 mediates long-range interactions that define the overall contours of the Xi super-structure.

To investigate how Xist RNA recruits SMCHD1, we screened the Xist interactome[21,55,57] and found that HNRNPK depletion abolished SMCHD1 recruitment (Fig. 7a, Supplementary Fig. 14b). Given that HNRNPK is also required for targeting PRC1 to the Xi[24,55,56], we examined the relationship of PRC1 to SMCHD1 localization. Strikingly, PRC1-depleted MEFs also failed to recruit SMCHD1 (Fig. 7a, Supplementary Fig. 14b). To determine if the PRC1 mark, H2AK119ub, is required for SMCHD1 recruitment, we treated cells with MG132, a proteasome inhibitor that depletes H2AK119ub by blocking the recycling of free ubiquitin (Supplementary Fig. 14c, d)[58]. Intriguingly, MG132 treatment also abolished SMCHD1 localization (Fig. 7b). Thus, it is likely that, beyond PRC1, the H2AK119ub mark itself is essential for SMCHD1 recruitment. In contrast, depleting EED, an essential PRC2 subunit, did not affect SMCHD1 targeting (Fig. 7a, Supplementary Fig. 14b), consistent with a human study[27]. Knocking down other Xist-interacting proteins—such as SPEN (Spen family transcriptional repressor), RBM15 (RNA-binding motif protein 15), or LBR (lamin B receptor)—did not affect SMCHD1 recruitment (Supplementary Fig. 14a, b). These data indicate that the effect of depleting HNRNPK, PRC1, and H2AK119ub on SMCHD1

localization is specific. Xist RNA therefore recruits SMCHD1 via a mechanism dependent on HNRNPK, PRC1, and H2AK119ub.

To examine the general relationship between PRC1 and SMCHD1, we examined their distribution on autosomes. As expected, the H3K27me3 mark was enriched near H2K119ub peaks on autosomes (Fig. 7c, Supplementary Fig. 14e). By contrast, SMCHD1 was not enriched at these autosomal Polycomb targets. We conclude that PRC1 and H2AK119ub may not be sufficient for SMCHD1 recruitment at autosomal sites. Thus, autosomal and Xi environments may be distinct.

Given a strong effect of the PRC1 pathway on SMCHD1 recruitment, we asked if its perturbation affects the merging of S1/S2 compartments, a key function of SMCHD1[28]. We performed in situ Hi-C on HNRNPK- and PRC1-KD cells in otherwise WT cells. Following PRC1 KD, S1/S2s did not become visible (Fig. 7d, e). Depleting HNRNPK also did not unveil S1/S2s (Fig. 7d, e). Rather, both conditions caused significantly sharpened megadomains and decreased inter-megadomain interactions (Fig. 7d–f). This finding is in line with a role for PRC1 in promoting long-range interactions that span the *Dxz4* boundary. The lack of re-emergent S1/S2 compartments despite a failure of SMCHD1 recruitment (Fig. 7a) is also consistent with PRC1 being necessary to form S1/S2s.

Collectively, these data demonstrate that Xist recruits PRC1 to partition the Xi into S1/S2 compartments (Figs. 5–7), and SMCHD1 is required to merge S1/S2s into the compartment-less Xi, even during Xi maintenance (Fig. 1). Moreover, the requirement of PRC1 in SMCHD1 recruitment necessitates S1/S2 compartments being formed prior to SMCHD1 localization, providing a potential mechanism that coordinates the stepwise folding process.

**Discussion**
Here we have provided insight into the stepwise folding mechanism of the Xi (Fig. 8a), a process we previously likened to "origami"[28]. Similar to origami, the folding and unfolding of the Xi appear to follow the "principle of reversibility" (unfolding taking the reverse pathway of folding). During de novo XCI, A/B compartments are first remodeled into S1/S2 structures. This step depends on Xist RNA (Fig. 5) and occurs as Xist spreads across the chromosome[28]. One of the factors recruited by Xist is PRC1 (Fig. 5f)[21,47–50,55]. We propose that, once recruited by Xist, the uneven distribution of PRC1 on the Xi aids in its phase separation into S1/S2 compartments, with Xist- and H2AK119-enriched chromatin partitioning into the S1 compartment (Figs. 6, 8b). Once PRC1 successfully re-compartmentalizes the Xi into the S1/S2 transitional state, SMCHD1 is recruited to merge S1/S2s to create a super-structure—the "one compartment" state that gives the appearance of being "compartment-less." In light of the complementary view of genome architecture offered by ligation-

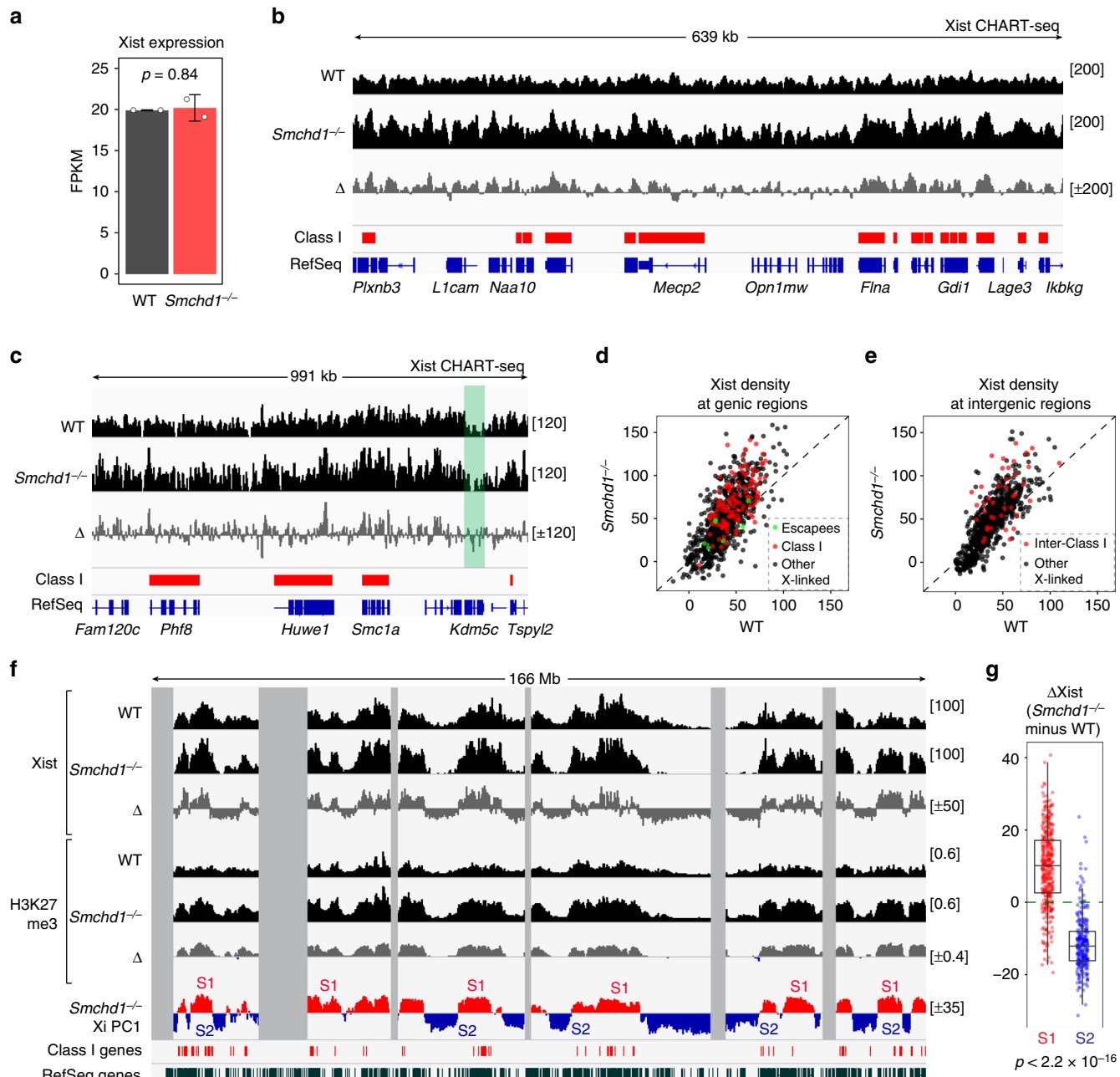

**Fig. 4** Ablating *Smchd1* traps Xist RNA in the S1 compartment. **a** RNA-sequencing (RNA-seq) fragments per kilobase of transcript per million mapped reads (FPKM) values of Xist in wild-type (WT) and *Smchd1*$^{-/-}$ mouse embryonic fibroblasts (MEFs). RNA-seq data from two WT and two *Smchd1*$^{-/-}$ clones treated with dimethyl sulfoxide (DMSO) were analyzed. *P* values are given by *t* test (unpaired, two-sided). Error bars, s.d. **b** Xist CHART (capture hybridization analysis of RNA target) profiles for a representative X-linked region harboring several Class I genes. Red bars, Class I genes. Scales shown in brackets. Δ, *Smchd1*$^{-/-}$ minus WT. **c** Xist CHART profiles for another representative X-linked region. Green-shaded area, an escapee (*Kdm5c*). **d** Xist density in gene bodies between WT (*x* axis) vs. *Smchd1*$^{-/-}$ (*y* axis) MEFs. Three categories of X-linked genes are shown. **e** Xist density in X-linked intergenic regions between WT (*x* axis) vs. *Smchd1*$^{-/-}$ (*y* axis) cells. Inter-Class I, intergenic regions flanked by two Class I genes. **f** Xist enrichment profiles across the X chromosome. Gray areas, unmappable regions. Also shown are H3K27me3 ChIP (chromatin immunoprecipitation followed by deep sequencing) profiles, the locations of Class I genes (red bars), and S1/S2 compartments in *Smchd1*$^{-/-}$ MEFs. **g** Box plots comparing the difference in Xist density between WT and *Smchd1*$^{-/-}$ cells of each 200-kb bin in S1 vs. S2 compartments. *P* values are given by the Wilcoxon's rank-sum test (unpaired, one-sided). Midline, median. Top and bottom of the box, first and third quartile. Whiskers, extension from the top or bottom to the furthest datum within 1.5 times the interquartile range

independent methods[59,60], we suggest that the Xi might not be completely compartment-less, as it might harbor an underlying organization not effectively captured by Hi-C.

Central to the stepwise folding mechanism is a requirement for PRC1 to either directly or indirectly enable SMCHD1 recruitment (Fig. 7), as also described in a recent work[41]. Here our perturbation studies reveal that architectural regulation is a key aspect of the relationship between PRC1 and SMCHD1. In the post-XCI state, ablating key components of the Xist-HNRNPK-PRC1-SMCHD1 pathway unravels the architectural layers of the Xi, one

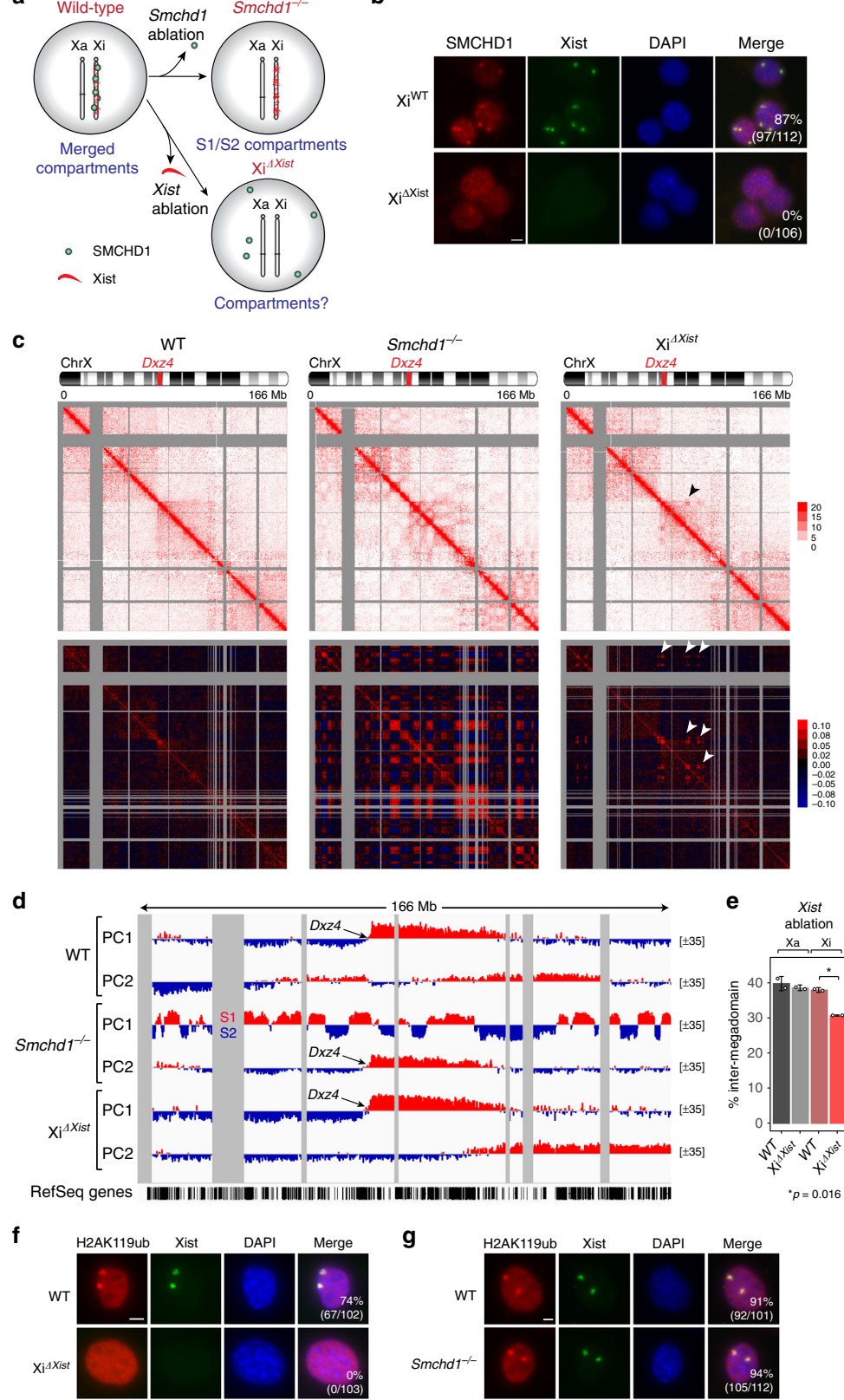

at a time, enabling the re-emergence of various hidden structures —in the reverse order of appearance during de novo XCI. Indeed, the transitional S1/S2 compartments reappear upon *Smchd1* ablation. Depleting PRC1 or HNRNPK in a SMCHD1-deficient background does not resurrect A/B compartments (Fig. 6), possibly because factors associated with active transcription may be

necessary to segregate A from B compartments[61]. Instead, we observed a disorganized and de-compartmentalized Xi, where S1/S2s are significantly weakened. In the wild-type background, depleting Xist, HNRNPK, or PRC1 also decompartmentalizes the Xi (Figs. 5, 7), although S1/S2 compartments may be resurrected transiently during this process. Interestingly, the aberrant Xi in

**Fig. 5** Ablating Xist RNA does not reveal S1/S2 compartments despite failed structural maintenance of chromosomes hinge domain containing 1 (SMCHD1) recruitment. **a** Schematic representation of the epigenetic status of the inactive X chromosome (Xi) in wild-type (WT), Smchd1[−/−], and Xi[ΔXist] fibroblasts. **b** Immuno-RNA-fluorescent in situ hybridization (Immuno-RNA-FISH) for SMCHD1 and Xist on WT [Xi[WT], 2lox(Xist+)] and Xi[ΔXist] fibroblasts. Scale bar, 10 μm. **c** Depth-corrected chromatin interaction maps of the Xi in WT, Smchd1[−/−], and Xi[ΔXist] fibroblasts binned at 200-kb resolution (top) and the corresponding Pearson's correlation maps (bottom). Gray-shaded areas, unmappable regions. Also see Supplementary Fig. 2a for active X chromosome (Xa) maps. Note that in the Xi maps of Xi[ΔXist] fibroblasts, the "super-loops" (arrowheads) formed by association between Xi regions exhibiting ATAC-seq (assay for transposase-accessible chromatin using sequencing) accessibility, BRG1 binding, cohesin binding, and topologically associated domain (TAD)-like structures not seen on the WT Xi can also be observed. Please see our reanalysis[68] of Hi-C data from the same Xi[ΔXist] fibroblasts[21] for in-depth description. **d** Principal component 1 (PC1) and PC2 values of the Xi. Gray-shaded areas, unmappable regions. Also see Supplementary Fig. 2b for PCs of the Xa. **e** Bar plots displaying the fraction of long-range interactions (>10 Mb) that span the megadomain boundary ("inter-megadomain" interactions) on the Xa and Xi in WT and Xi[ΔXist] fibroblasts. Two replicates were analyzed, with P-values determined by the t test (unpaired, one-sided). Error bars, s.d. **f** Immuno-RNA-FISH for H2AK119ub and Xist on WT [2lox(Xist+)] and Xi[ΔXist] fibroblasts. Number of cells with Xist clouds and co-localizing H2AK119ub foci is shown. Scale bar, 5 μm. **g** Immuno-RNA-FISH for H2AK119ub and Xist on WT and Smchd1[−/−] MEFs. Number of cells with H2AK119ub foci co-localizing with Xist clouds is shown. Scale bar, 5 μm

these cells exhibits stronger insulation between the two megadomains. We speculate that PRC1's known self-associating and/or chromatin-compacting properties not only facilitate short-range interactions through condensation but also long-range interactions across the *Dxz4* border (Fig. 8b), explaining why its loss sharpens the megadomain boundary. Consistently, deleting Repeat B, the RNA motif required for Xist to interact with HNRNPK and to recruit PRC1, also causes a disorganized Xi and reduces inter-megadomain interaction[24]. In summary, the antagonistic role of SMCHD1 on PRC1-mediated Xi compartments and the obscuring effect of PRC1 on megadomains illustrate that the Xi is shaped by multiple, sometimes competing folding mechanisms (Fig. 8c). Thus, the Xi is far from the "structure-less" body that it was once thought to be.

A role for PRC1 in S1/S2 compartmentalization supports a mechanism involving LLPS, an idea widely proposed to explain chromosome compartmentalization. A major gap has been the lack of evidence for critical chromatin factors with known phase-separating property. In this regard, a role for PRC1 is especially attractive, given recent evidence that PRC1 forms liquid droplets in vitro and nuclear condensates in vivo that are sensitive to hexanediol[12,13]. Our data indicating that depleting PRC1 abolishes large-scale S1/S2 compartments lends direct support to the hypothesis that LLPS contributes to chromosome compartmentalization.

Our work also demonstrates a clear functional relevance for S1/S2 compartments. Indeed, the S1/S2 organization is important for both global and regional Xist spreading, and associated gene silencing. During de novo XCI, failure to merge S1/S2s compromises regional Xist spreading, resulting in eroded H3K27me3 domains and impaired gene silencing, specifically at regions harboring SMCHD1-sensitive "Class I genes"[28]. By contrast, in post-XCI cells, depleting SMCHD1 did not affect regional spreading, suggesting that once Xist establishes an epigenetic memory in Class I regions, SMCHD1 becomes dispensable for local Xist spreading. An Xist-induced epigenetic memory is in line with earlier observations[43,50,62]. While regional Xist spreading is preserved in post-XCI cells, ablating *Smchd1* affects global Xist distribution. Spatial segregation between S1 and S2 traps Xist RNA in the S1 compartment where it is produced, rendering the RNA unable to spread as efficiently into adjoining S2 structures. As a consequence, H3K27me3 marks accumulate in S1 domains (Fig. 3g, h). H2AK119ub marks also accumulate in S1 (Fig. 6a, b).

In addition to merging S1/S2 compartments, SMCHD1 also attenuates Xi TADs. Ablating *Smchd1* strengthens Xi TADs both during de novo XCI[28] and in the maintenance phase (Supplementary Fig. 4). Notably, TADs on the Smchd1[−/−] Xi remain weaker than those on the Xa, indicating the existence of SMCHD1-independent mechanisms. Inefficient cohesin loading

and sliding due to attenuated transcription[63], impaired CTCF binding by DNA methylation[64], and other architectural proteins at play on the Xi are candidate mechanisms to be explored in the future.

Altogether, perturbing SMCHD1 and S1/S2 structures result in a weakened silent state that is more easily disrupted by pharmacological DNA demethylation (Fig. 2). Our findings support a synergy between SMCHD1 and DNA methylation contributes to Xi maintenance. The heightened reactivation efficiency in Smchd1[−/−] cells seems at odds with elevated repressive histone marks on the Xi (Figs. 3, 6). As reawakening the healthy allele of disease genes on the Xi presents a therapeutic opportunity in X-linked disorders[38], our finding suggests dual targeting of SMCHD1 and DNA methylation as a candidate approach in developing such treatment strategy.

Overall, our work complements several recent studies on the role of SMCHD1 in XCI[28,35,36,41,65]. There are, however, several notable differences. For instance, one study indicated that ablating *Smchd1* in post-XCI cells caused a reversion to "A/B" compartments similar to those on the Xa[36], rather than an unveiling of the S1/S2 compartments we describe. There are several reasons to believe that the compartments described by Gdula et al.[36] may be more akin to S1/S2s. First, they are larger than classical A/B compartments and are partitioned along the Xi like S1/S2s. Second, the partitioning follows an S1/S2 pattern that is distinguished by Xist, H3K27me3, and H2AK119ub densities, rather than the transcriptional states that typify A/B compartments. Third, classical A/B compartments do not require Xist RNA, HNRNPK, and PRC1, but the structures we describe clearly do. Thus, ablating *Smchd1* results in compartment structures (S1/S2) that have properties that distinguish them from classical A/B compartments. A second notable difference regards the enrichment of H3K27me3 following SMCHD1 depletion. Another group has reported that ablating *Smchd1* in post-XCI cells also causes an overall increase of H3K27me3 on the Xi[65]. However, we further linked the preferential accumulation of H3K27me3 on the S1 domains to altered distribution of Xist RNA and Xi topology. Altogether, these works highlight an exciting area of 3D chromosome biology and underscores the utility of the Xi for dissecting structure–function relationships.

## Methods

**Cell lines**. The clonal F1 hybrid MEF line (EY.T4), Xi[ΔXist] fibroblasts (clone III.20), 2lox(Xist+) fibroblasts (clone III.8), and a MEF line of pure *Mus castaneous* background have been described[29,44,66]. Cells were grown in MEF media [Dulbecco's modified Eagle's medium, high glucose, GlutaMAX supplement, pyruvate (10569044, Thermo Fisher Scientific), 10% fetal bovine serum, 25 mM HEPES, pH 7.2–7.5 (15630130, Thermo Fisher Scientific), 1× MEM non-essential amino acids (11140076, Thermo Fisher Scientific), 1× penicillin-streptomycin (15140163, Thermo Fisher Scientific), and 0.1 mM β-mercaptoethanol (21985023, Thermo

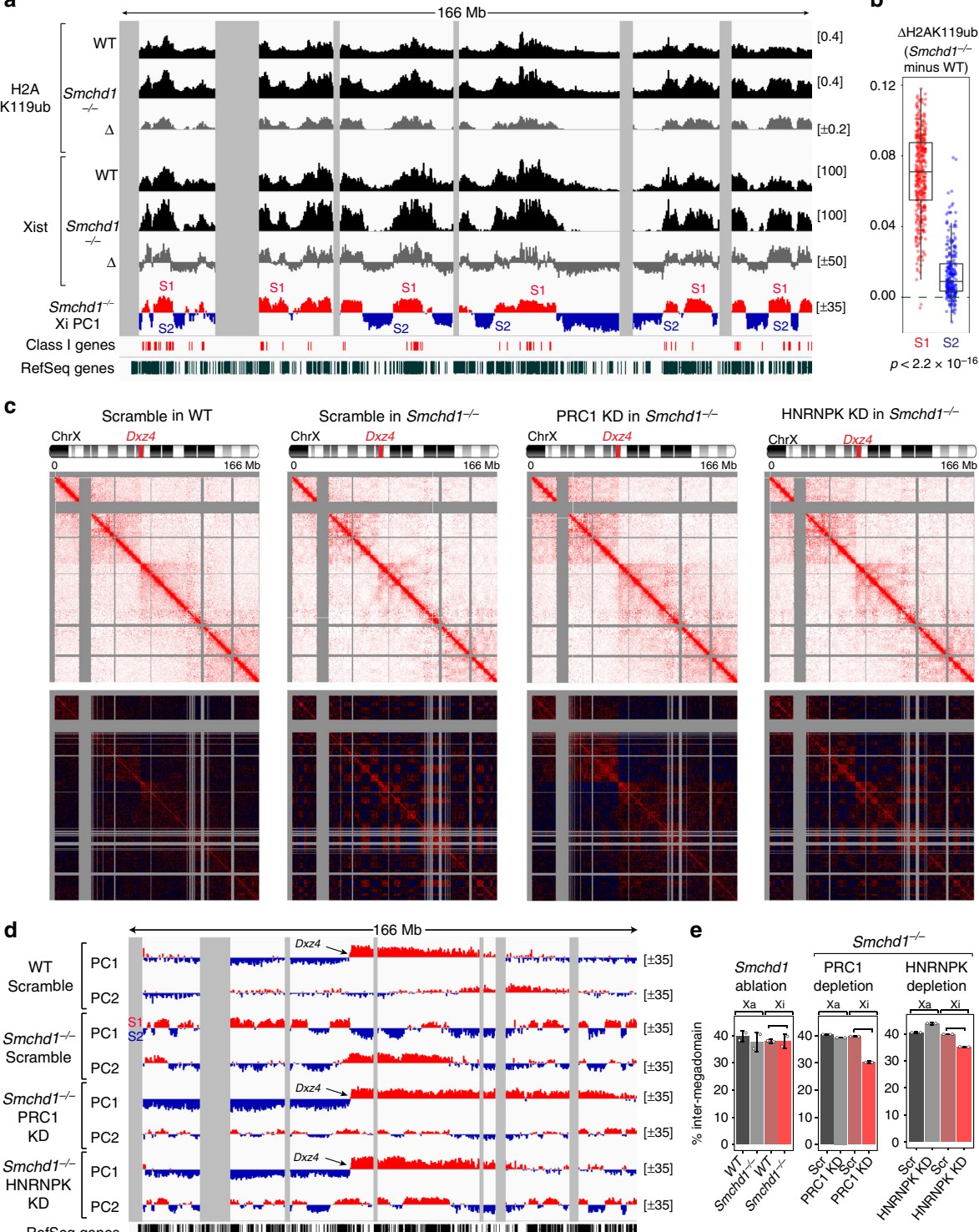

Fisher Scientific)]. *Smchd1*[-/-] MEF lines were generated by transfecting the EY.T4 line using Lipofectamine LTX and Plus Reagent (15338100, Thermo Fisher Scientific) with plasmids (PX461; 48140, Addgene) expressing Cas9 D10A nickase, enhanced green fluorescent protein (EGFP), and a pair of guide RNAs targeting the first exon of *Smchd1*, as described previously[28]. To control off-target effects, two different pairs of guide RNAs were used (pair 1: GGGAGAGATGGCGCCGT CGA, CGAGAGGCCGGCGGGATCGC; pair 2: CTTGTTTGACCGGCGCGGGA, CACCGTCCTACAGCCGTCGA) (see Supplementary Data 3 for oligos used for cloning). WT control MEF lines were generated by transfecting EY.T4 with the PX461 plasmid without guide RNAs. At 24 h after transfection, cells were dissociated and transfected cells were sorted by EGFP into 96-well plates, with one cell plated in each well. MEF clones were screened by western blot using an antibody

against SMCHD1 (Sigma, HPA039441). To confirm the presence of frame-shifting mutations, the CRISPR-targeted region was PCR amplified from genomic DNA (primers are listed in Supplementary Data 3) and subsequently cloned into a TOPO-TA vector (450030, Thermo Fisher Scientific) for Sanger sequencing. For RNA-seq, two WT and two *Smchd1*[-/-] clones were used. For in situ Hi-C, ChIP-seq, and CHART-seq, WT clone 1 and *Smchd1*[-/-] clone 1 were used.

For Aza (A3656, Sigma) treatment, we incubated WT and *Smchd1*[-/-] MEFs with media containing either DMSO or Aza (0.3 μM) for 5 days. Media supplemented with DMSO or Aza were changed daily.

For MG132 (M7449, Sigma) treatment, 0.1 million WT MEFs were plated in one well of a 12-well plate (or 0.5 million in one well of a 6-well plate). The next day, media were replaced with fresh media containing 10 μM MG132. After

**Fig. 6** Depleting polycomb repressive complex 1 (PRC1) or heterogeneous nuclear ribonucleoprotein K (HNRNPK) disrupts S1/S2 compartments in *Smchd1⁻/⁻* mouse embryonic fibroblasts (MEFs). **a** H2AK119ub enrichment profiles across the X chromosome. Gray areas, unmappable regions. Also shown are Xist CHART (capture hybridization analysis of RNA target) profiles, the locations of Class I genes (red bars), and S1/S2 compartments in *Smchd1⁻/⁻* MEFs. Δ, *Smchd1⁻/⁻* minus wild type (WT). Shown are "comp" tracks (compiled from all reads) of H2AK119ub ChIP-seq (chromatin immunoprecipitation followed by deep sequencing). Allele-specific tracks have been deposited to Gene Expression Omnibus (GEO) (GSE116413). **b** Box plots comparing the difference in H2AK119ub density between WT and *Smchd1⁻/⁻* cells of each 200-kb bin in S1 vs. S2 compartments. *P* values are given by the Wilcoxon's rank-sum test (unpaired, one-sided). Midline, median. Top and bottom of the box, first and third quartile. Whiskers, extension from the top or bottom to the furthest datum within 1.5 times the interquartile range. **c** Depth-corrected chromatin interaction maps of the inactive X chromosome (Xi) in WT MEFs treated with control (Scramble) small interfering RNA (siRNA), and *Smchd1⁻/⁻* MEFs treated with control, RING1A/RING1B (PRC1 KD), or HNRNPK (HNRNPK KD) siRNA binned at 200-kb resolution (top) and the corresponding Pearson's correlation maps (bottom). Gray-shaded areas, unmappable regions. Also see Supplementary Fig. 13a for the active X chromosome (Xa) maps. **d** Principal component 1 (PC1) and PC2 values of the Xi. Regions with positive PC1 values represent the S1 compartment or the telomeric megadomain. Gray-shaded areas, unmappable regions. Also see Supplementary Fig. 13b for PCs of the Xa. **e** Bar plots displaying the fraction of long-range interactions (>10 Mb) that span the *Dxz4* megadomain boundary ("inter-megadomain" interactions) under different conditions. Two replicates were analyzed, with *P* values determined by the *t* test (unpaired, one-sided). Error bars, s.d.

incubation for 6 h, cells were harvested for immuno-RNA-fluorescent in situ hybridization (immuno-RNA-FISH) and western blot analyses.

**RNA-FISH and immunostaining**. Fibroblasts were grown overnight on glass coverslips coated with 0.2% gelatin. The next day, cells were rinsed in chilled phosphate-buffered saline (PBS) for 5 min three times, pre-extracted with CSKT buffer (100 mM NaCl, 300 mM sucrose, 10 mM PIPES, 3 mM MgCl₂, 0.5% Triton X-100, pH 6.8) supplemented with 10 mM ribonucleoside vanadyl complex (VRC; S1042S, NEB) for 10 min on ice, and fixed with 4% paraformaldehyde (15713, Electron Microscopy Sciences) in PBS at room temperature for 10 min (for H2AK119ub and H3K27me3) or chilled 100% methanol at −20 °C for 10 min (for SMCHD1). After fixation, cells were washed with chilled PBS for 5 min, further permeabilized by CSKT with 10 mM VRC for 10 min on ice, followed by three washes with chilled PBS for 5 min. After blocking with 5% normal goat serum (PCN5000, Thermo Fisher Scientific) with 0.6 U/μl Protector RNase Inhibitor (3335402001, Sigma) in PBS at room temperature for 1 h, immunostaining were performed using rabbit monoclonal H3K27me3 antibodies (GTX60892, GeneTex, 1:3000), rabbit monoclonal H2AK119ub antibodies (8240S, Cell Signaling, 1:1000), or rabbit polyclonal SMCHD1 antibodies (HPA039441, Sigma, 1:200) in antibody dilution buffer (5% normal goat serum, 0.6 U/μl Protector RNase Inhibitor, and 0.2% Tween-20 in PBS) at room temperature for 1 h. Following primary antibody incubation, cells were washed with PBST (PBS with 0.2% Tween-20) for 5 min three times, and incubated with secondary antibody at room temperature for 1 h. Three 5-min PBST washes were then performed, followed by re-fixation with 4% paraformaldehyde (15713, Electron Microscopy Sciences) in PBS at room temperature for 10 min and two 5-min washes in PBS. Cells were then dehydrated by sequential incubation with 70%, 80%, 90%, and 100% ethanol for 3 min and air-dried. RNA-FISH was performed following immunostaining with Xist probes [40 ng of nick-translated pSx9-3 plasmids or 2 ng Xist oligonucleotide probes (see Supplementary Data 3 for sequences of FISH probes)[67]]. Images were acquired with a Nikon Eclipse 90i microscope and a Hamamatsu CCD camera. Image analysis was performed using Volocity (Perkin-Elmer).

**In situ Hi-C**. In situ Hi-C was performed following published protocols[6,28] using *Hin*dIII (100 U/μl; R0104M, NEB) and biotin-14-dATP (19524016, Thermo Fisher Scientific) on one WT and one *Smchd1⁻/⁻* MEF clone, either without treatment, or transfected with scramble or HNRNPK- or RING1A- and RING1B-targeting small interfering RNA (siRNA) (see below for details of siRNA KD). We also performed in situ Hi-C on Xi^ΔXist fibroblasts. Hi-C libraries were sequenced on Illumina HiSeq, generating 60–150 millions paired-end 50 nucleotide reads per sample.

**In situ Hi-C analysis**. Each end of the Hi-C read pairs were aligned separately as single-end reads to the 129S1/SvJm (mus) and CAST/Eih (cas) genome using NovoAlign, followed by assignment of the allelic origin[28]. Alignments of the two ends were then joined. Allele-specific Hi-C contact matrices were constructed using Homer. Iterative correction of Hi-C matrices and construction of Pearson's correlation matrices at 200-kb resolution were performed using HiTC. We noticed that the intensity of Pearson's correlation maps is influenced by sequencing depth. Therefore, we randomly down-sampled Hi-C data to match the sequencing depth. Depth-corrected Hi-C contact maps and Pearson's correlation maps at 200-kb resolution were displayed in the manuscript. Unmappable regions, defined as bins with low read coverage (bottom ~14%), were set to NA before iterative correction. PCA was performed on Hi-C data without downsampling using Homer at 200-kb resolution, and the principal component profiles were visualized using IGV or R. To estimate the number and the average size of chromosome compartments, compartments were called using Homer at 100-kb resolution in combination with custom R script. We noted that the replicate 2 of in situ Hi-C in Xi^ΔXist fibroblasts

have more PCR duplicates than other samples. Therefore, this dataset was not further down-sampled and only used to calculate inter-megadomain interactions.

To calculate the correlation between compartment profiles and various chromatin features presented in Fig. 1c, we binned the X chromosome at 200 kb and computed Pearson's correlation coefficients, with unmappable regions excluded. H3K4me3 ChIP-seq (GSE33823), H2AK119ub ChIP-seq (GSE107217), H3K27me3 ChIP-seq (GSE36905), Xist CHART-seq (GSE48649), and SMCHD1 DamID-seq (GSE99991) datasets in the EY.T4 WT MEF line were downloaded from Gene Expression Omnibus (GEO). For Xist CHART-seq, H3K27me3 ChIP-seq, and H2AK119ub ChIP-seq, we used input-subtracted coverage profiles generated by SPP with smoothing using 1-kb windows recorded every 500 bp[43].

Insulation scores were generated from iteratively corrected 100-kb binned Hi-C matrices using the script matrix2insulation.pl (publically available on Github: https://github.com/dekkerlab/giorgetti-nature-2016)[22] with parameters −is 500000 −ids 400000 −im iqrMean −nt 0 −ss 200000. For TADs and TAD boundaries, we used the coordinates identified from mouse embryonic stem cells defined previously[2]. To investigate the relationships between S1/S2 compartments and TAD boundaries, we compared the PC1 values of *Smchd1⁻/⁻* MEF Xi computed at 100-kb resolution with insulation profiles generated at the same resolution.

To compare the compartment profiles of MEFs with those of cells undergoing de novo XCI, we analyzed previously published Hi-C data of undifferentiated female ES cells (D0 ES) and embryoid bodies (EBs) formed after 4 or 7 days of differentiation (D4 and D7 EB) (GSE99991)[28]. We performed PCA on these datasets at 200-kb resolution using Homer. The hierarchical clustering analyses using the Euclidean or Pearson's correlation distance on compartment profiles were performed using the hclust function in R.

To quantify the strength of megadomains, we computed %inter-megadomain interactions, defined as the fraction of long-range chromatin interactions (>10 Mb, approximately twice the size of the largest TAD on the X chromosome) that span the megadomain boundary (73 Mb, mm9) on depth-corrected Hi-C contact maps at 200-kb resolution. When biological replicates were available, the fraction of inter-megadomain interactions in two biological replicates were computed separately and presented the means and standard deviation as bar plots. For the inter-megadomain interactions of WT MEFs plotted in Figs. 5e, 6e, and 7f, Hi-C data of WT MEF clone without treatment and WT MEFs transfected with scramble siRNA were considered as biological replicates. For the inter-megadomain interactions of *Smchd1⁻/⁻* MEFs plotted in the left panel of Fig. 6e, Hi-C data of *Smchd1⁻/⁻* MEFs without treatment and *Smchd1⁻/⁻* MEFs transfected with scramble siRNA (replicate 1) were considered as biological replicates.

**RNA-sequencing**. Total RNA was extracted from two WT and two *Smchd1⁻/⁻* clones treated with either DMSO or Aza using TRIzol (15596018, Thermo Fisher Scientific). Following ribosomal RNA depletion using RiboMinus Eukaryote Kit v2 (A15020, Thermo Fisher Scientific), strand-specific RNA-seq libraries were prepared[28]. All libraries were sequenced with Illumina HiSeq, generating 32–63 millions paired-end 50 nucleotide reads per sample.

**RNA-seq analysis**. RNA-seq reads were aligned to the 129S1/SvJm (mus) and CAST/Eih (cas) genome using Tophat2, followed by assignment of the allelic origin[28]. After removing PCR duplicates, unique exonic reads mapped to each gene were quantified by Homer. To compare the fold changes of autosomal and X-linked genes, we analyzed only genes with fragments per kilobase of transcript per million mapped exonic reads (FPKM) >0.1. RNA-seq data of neural progenitor cells derived from WT or *Smchd1⁻/⁻* female mouse embryonic stem cells (GSE99991)[28] were downloaded from GEO. For allele-specific analysis, we defined %mus as the percentage of mus-specific exonic reads in all allele-specific (mus-specific + cas-specific) exonic reads of each transcript. To classify X-linked genes,

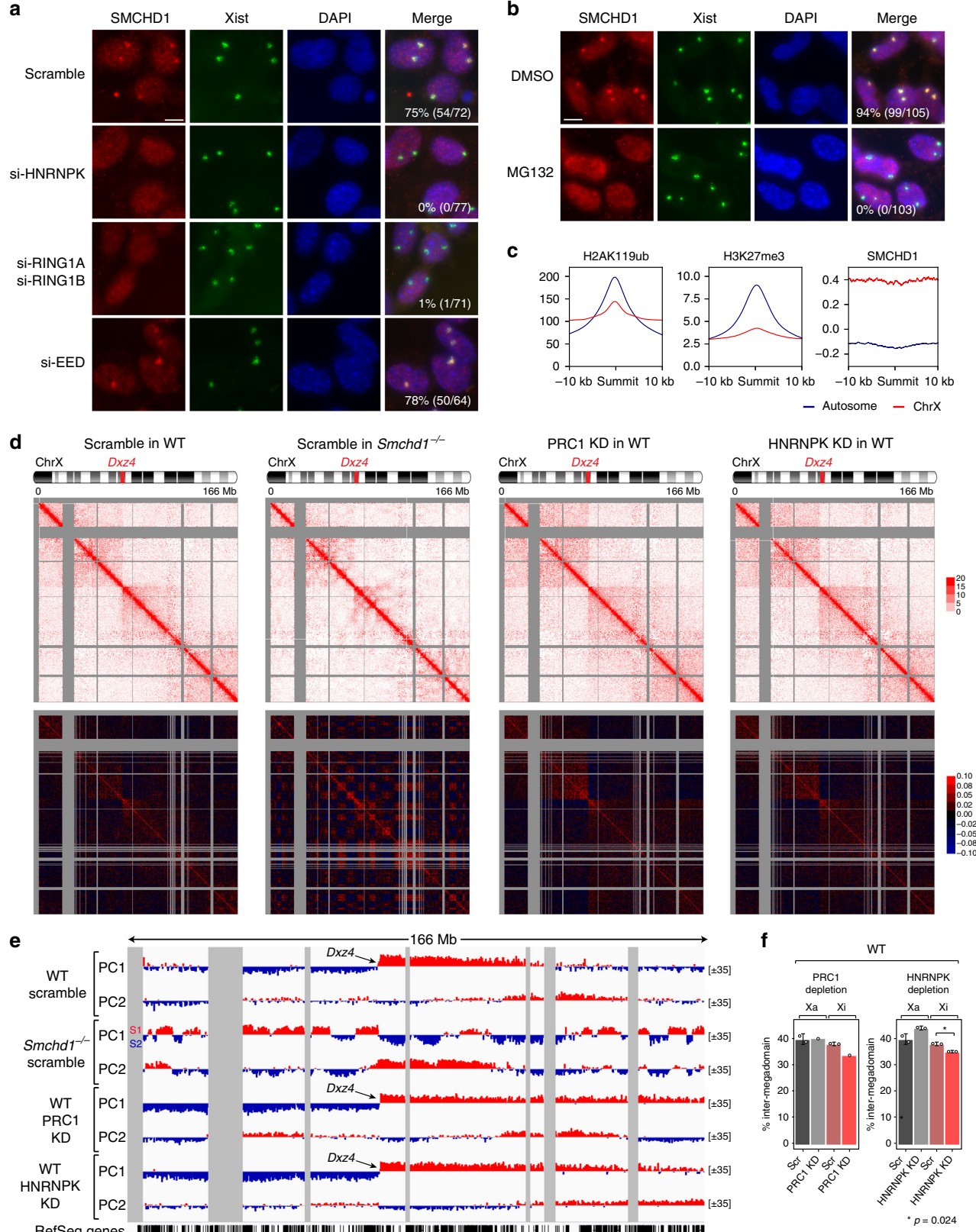

we defined active genes as genes with non-zero FPKM in all samples. Allele-assessable genes were defined as active genes with at least 13 allele-specific reads in all samples. A small fraction of genes overlap with incorrectly annotated single-nucleotide polymorphisms (SNPs) and produce unexpected allelic skewing, which were identified by analyzing the RNA-seq datasets of tail-tip fibroblasts (TTF) (GSE58524) and MEFs of pure *Mus castaneous* background[66]. Allele-assessable genes having %mus >9.09% in either pure cas TTFs or MEFs were considered as

genes with miscalled SNPs. After excluding these genes, we defined escapees as active and allele-assessable genes whose expression from the Xi is >10% of the expression from the Xa in at least one WT MEF clone. Genes subject to X-inactivation were defined as active and allele-assessable genes that were not genes with miscalled SNPs or escapees. Reactivated genes were defined as genes subject to XCI whose average %mus in two *Smchd1*[−/−] clones treated with Aza was (1) 3-fold greater and (2) greater with the associated *p* value <0.1 (one-sided *t* test) than the

**Fig. 7** Heterogeneous nuclear ribonucleoprotein K (HNRNPK) and polycomb repressive complex 1 (PRC1) mediate structural maintenance of chromosomes hinge domain containing 1 (SMCHD1) recruitment and the stepwise folding of the inactive X chromosome (Xi). **a** Immuno-RNA-fluorescent in situ hybridization (immuno-RNA-FISH) for SMCHD1 and Xist on female wild-type mouse embryonic fibroblasts (WT MEFs) treated with small interfering RNA (siRNA) targeting HNRNPK, PRC1, and PRC2. Number of cells with SMCHD1 foci co-localizing with Xist clouds in a representative biological replicate is shown. Scale bar, 10 μm. **b** Immuno-RNA-FISH for SMCHD1 and Xist on female WT MEFs treated with dimethyl sulfoxide (DMSO) or MG132. Number of cells with SMCHD1 foci co-localizing with Xist clouds in a representative biological replicate is shown. Scale bar, 10 μm. **c** Chromatin binding profiles of H2AK119ub (GSE107217), H3K27me3 (GSE33823), and SMCHD1 (GSE99991) in female wild-type MEFs averaged over each 20-kb region centered at the summit of a H2AK119ub peak. **d** Depth-corrected chromatin interaction maps of the Xi in WT MEFs treated with control (scramble) siRNA, $Smchd1^{-/-}$ MEFs treated with control siRNA, and WT MEFs treated with RING1A/RING1B (PRC1 knockdown (KD)) or HNRNPK (HNRNPK KD) siRNA binned at 200-kb resolution (top) and the corresponding Pearson's correlation maps (bottom). Gray-shaded areas, unmappable regions. Also see Supplementary Fig. 13a for the active X chromosome (Xa) maps. **e** Principal component 1 (PC1) and PC2 values of the Xi. Regions with positive PC1 values represent the S1 compartment or the telomeric megadomain. Gray-shaded areas, unmappable regions. Also see Supplementary Fig. 13b for PCs of the Xa. **f** Bar plots displaying the fraction of long-range interactions (>10 Mb) that span the $Dxz4$ megadomain boundary ("inter-megadomain" interactions) under different conditions. For HNRNPK KD, two replicates were analyzed, with $P$ values determined by the $t$ test (unpaired, one-sided). Error bars, s.d.

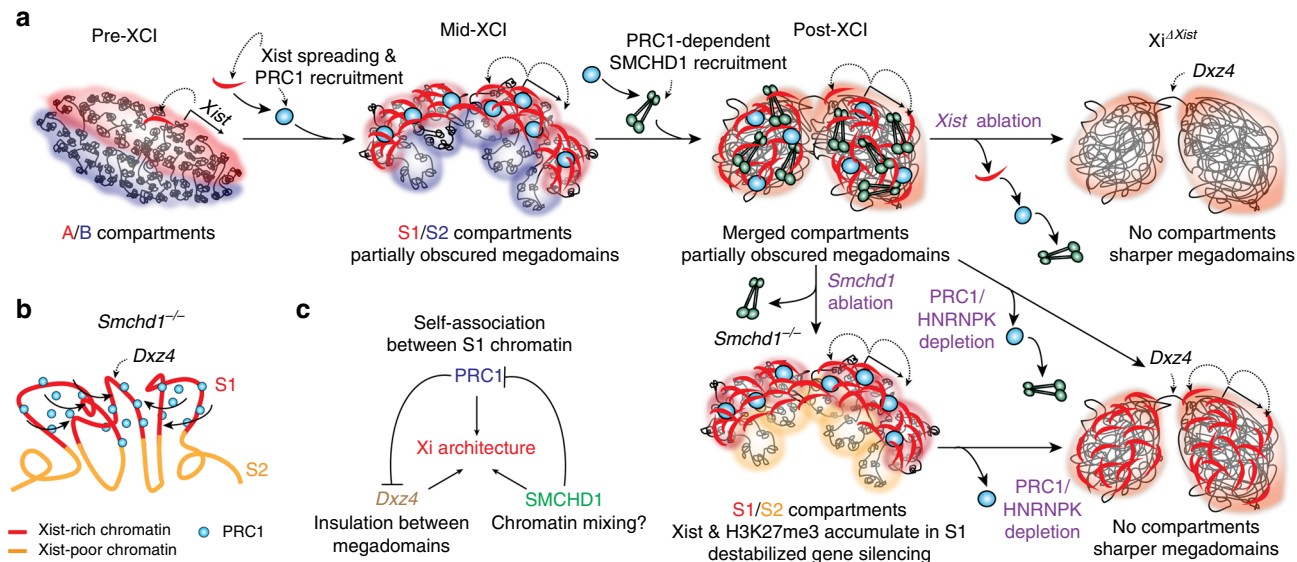

**Fig. 8** A model for the folding and unfolding of the inactive X chromosome (Xi) origami. **a** Summary: A stepwise folding process for the transformation of the Xi ("Origami Model"). Xist RNA is produced from the A compartment. Following transcription, Xist initially spreads to co-segregated A-compartmental chromatin. Through recruiting polycomb repressive complex 1 (PRC1), Xist reconfigures the Xi into S1/S2 compartments. Following structural maintenance of chromosomes hinge domain containing 1 (SMCHD1) recruitment (via a mechanism requiring heterogeneous nuclear ribonucleoprotein K (HNRNPK), PRC1, and H2AK119ub), S1/S2 compartments are merged to form a compartment-less structure. In cells losing SMCHD1 after completing XCI, S1/S2 compartments reappear, coinciding with accumulation of Xist, H3K27me3, and H2AK119ub in the S1 compartment and destabilized gene silencing. Depleting PRC1 or HNRNPK diminishes the reappeared S1/S2 compartments, resulting in a "de-compartmentalized" Xi with sharper partition between the two megadomains. In cells that have undergone Xist ablation post XCI, both SMCHD1 and PRC1 fail to be recruited to the Xi, also leading to a "de-compartmentalized" Xi with sharper megadomains. **b** Model: In the absence of SMCHD1, Xist-rich chromatin co-segregates, likely via PRC1 self-association, leading to the formation of S1/S2 compartments. The self-association between S1 chromatin creates long-range interaction across the $Dxz4$ boundary, thus partially obscuring megadomains. **c** Three major folding mechanisms that define the large-scale Xi architecture: self-association between Xist/H2AK119ub-enriched chromatin to form S1/S2 compartments through PRC1, potential chromatin mixing activity of SMCHD1, and inter-megadomain insulation by $Dxz4$. When superimposed on the Xi, these folding mechanisms sometimes interfere with each other—SMCHD1 attenuates S1/S2 structure, and PRC1 obscures megadomains

average %mus in two WT clones treated with DMSO. To compare the cumulative distribution of %mus of genes subject to XCI, we averaged the %mus of two clones. To compare the effect of combined $Smchd1$ ablation and Aza treatment on genes in S1 vs. S2 compartments, we defined the associated PC1 value of each X-linked gene as the PC1 value of the bin containing its transcriptional start site. Genes associated with positive PC1 values on the $Smchd1^{-/-}$ Xi (100-kb resolution) were considered as residing in the S1 compartment. To investigate the effect of $Smchd1$ ablation and/or Aza treatment on the expression of escapees, we compared the %mus of 11 genes defined as escapees in both WT clones.

We previously defined three classes (Class I–III) of genes subject to XCI[28]. Class I genes require SMCHD1 to be silenced during de novo XCI, and their gene bodies become depleted of the H3K27me3 mark in the absence of SMCHD1. Class II genes remain silent, enriched with H3K27me3, but exhibit aberrant H3K4me3 peaks at the promoter on the Xi. Class III genes are resistant to $Smchd1$ ablation, with both the expression and the chromatin state (H3K4me3 and H3K27me3) unchanged in cells lacking SMCHD1. To investigate the difference between

different classes of Xi genes in response to post-XCI $Smchd1$ ablation and Aza treatment, we intersected the 126 Class I genes with the 216 genes subject to XCI in MEFs. We then compared the cumulative distribution of the resulting 79 Class I genes and other 137 non-Class I genes. We noted that among the 216 genes subject to XCI, three genes ($Rgag4$, $Timp1$, and $Slc10a3$) were excluded from the categorization analysis in Wang et al.[28], because they overlap with a longer gene. We assigned these three genes as "Class I" ($Slc10a3$), "Class III" ($Rgag4$), and "Unclassified" ($Timp1$), respectively, based on the criteria described previously[28].

To visualize RNA-seq coverage, we generated strand-resolved fpm-normalized bigWig files from the raw RNA-seq reads for all reads (comp), mus-specific (mus) reads, and cas-specific (cas) reads separately, which were displayed using IGV with scales indicated in each track.

**ChIP-seq of H3K27me3 and H2AK119ub.** WT and $Smchd1^{-/-}$ MEFs were cross-linked in PBS with 1% formaldehyde at room temperature for 10 min at 2 million

cells/ml and quenched with 0.125 M glycine at room temperature for 5 min. Cross-linked cells were washed three times with chilled PBST (1× PBS with 0.05% Tween-20) before snap-frozen in liquid nitrogen. Five million cross-linked cells were thaw on ice for 15 min, resuspended in 2 ml Buffer 1 [50 mM HEPES, pH 7.5, 150 mM NaCl, 1 mM EDTA, 0.5% Nonidet P-40, 0.2% Triton X-100, 1× cOmplete EDTA-free Protease Inhibitor Cocktail (PIC; 11873580001, Sigma)], and rotated at 4 °C for 10 min. Nuclei were pelleted by centrifugation at $1700 \times g$ for 5 min at 4 °C, resuspended with 2 ml Buffer 2 (10 mM Tris, pH 8.0, 200 mM NaCl, 5 mM EDTA, 2.5 mM EGTA, 1× PIC) supplemented with 40 µl Purelink RNaseA (12091021, Thermo Fisher Scientific), and incubated for 15 min with rotation at 4 °C. Nuclei were then pelleted again by centrifugation at $1700 \times g$ for 5 min at 4 °C, resuspended with 1.5 ml Buffer 3 (10 mM Tris, pH 8.0, 5 mM EDTA, 2.5 mM EGTA, 1× PIC) supplemented with 75 µl 10% N-lauroylsarcosine, and incubated for 15 min with rotation at 4 °C. Following incubation, nuclei were transferred to five Evergreen Scientific polystyrene 1.5 ml tubes (250 µl nuclear suspension per tube) and sonicated for 10 min using Qsonica Q800R3 (20 s ON/20 s OFF; amplitude 50%). Sonicated chromatin was mixed at 1:1 ratio with 2× immunoprecipitation (IP) buffer (30 mM Tris, pH 8.0, 300 mM NaCl, 2% Triton X-100, 0.5% N-lauroylsarcosine, 1× PIC) and centrifuged at full speed for 30 min at 4 °C to remove insoluble debris. Two micrograms of chromatin was set aside as the input sample. For ChIP, 20 µg chromatin was mixed with antibodies overnight with rotation at 4 °C. Antibodies used were: α-H3K27me3 (2 µl; Millipore 07-449), α-H2AK119ub (2 µl; Cell Signaling, 2899S), and normal rabbit IgG (2729S, Fisher Scientific), which served as a negative control. Following overnight incubation, chromatin was mixed with 20 µl Dynabeads Protein G (10004D, Thermo Fisher Scientific), which had been washed three times in 1× IP buffer (15 mM Tris, pH 8.0, 150 mM NaCl, 1% Triton X-100, 0.25% N-lauroylsarcosine, 2.5 mM EDTA, 1.25 mM EGTA, 1× PIC), and incubated for 2 h with rotation at 4 °C. Beads were then washed with 1 ml 1× IP buffer once, 1 ml wash buffer [50 mM HEPES, pH 7.5, 500 mM LiCl, 1 mM EDTA, 1% Nonidet P-40, 0.7% sodium deoxycholate, 1× PIC] six times, and 1× TEN buffer (10 mM Tris, pH 8.0, 50 mM NaCl, 1 mM EDTA) once. Following elution, crosslinks reversal, and validating ChIP quality, input and ChIP-enriched DNA were subject to library preparation using NEBNext ChIP-Seq Library Prep Master Mix Set for Illumina (E6240S, NEB) and sequenced on Illumina HiSeq, generating ~30 millions paired-end 50 nucleotide reads per sample. H3K27me3 ChIP-seq was performed in two biological replicates. Some of the samples were sequenced at the rapid-run mode on Illumina HiSeq to generate pair-end 69 nucleotide reads, which were trimmed to 50 nucleotides before downstream analysis.

### Analysis of H3K27me3 and H2AK119ub ChIP-seq datasets.

Input and ChIP-seq reads were aligned to the 129S1/SvJm (mus) and CAST/Eih (cas) genome using NovoAlign, followed by assignment of the allelic origin[28]. After removing PCR duplicates, we generated fpm-normalized bigWig files from ChIP-seq reads for all uniquely mapped (comp), mus-specific (mus), and cas-specific (cas) reads separately, which were displayed using IGV with scales indicated in each track. As these two histone marks on the X chromosome originate predominantly from the Xi, we displayed and used the comp tracks for analyses in the manuscript. The ΔH2A-K119ub and ΔH3K27me3 tracks (Smchd1$^{-/-}$ minus WT) were computed by the bigwigCompare function of deepTools. H3K27me3 density at each X-linked genic and intergenic regions was computed by Homer, with the categorization of intergenic regions defined previously[28]. In addition to fpm normalization, we also generated input-subtracted ChIP tracks by SPP, with smoothing using 1-kb windows recorded every 500 bp, and fold-enrichment profiles of ChIP-seq over input using MACS2.

To compare the distribution of H2AK119ub and SMCHD1, we first used MACS2 ($q = 0.01$) to call summits of H2AK119ub. After filtering out peaks that mapped to ribosomal DNA and a false-positive region on chromosome 2 (chr2: 98,506,741–98,507,176, mm9), the coverage of H2AK119ub ChIP-seq (GSE107217), H3K27me3 ChIP-seq (GSE48649_MEF.K27me3.comp.bedGraph. gz), and SMCHD1 DamID-seq (GSM3036552_SMCHD1.rep1.DamID.comp.bw), at a 20-kb region centered at each summit were computed and plotted as average profiles and heat maps by deeptools (v3.1.2).

### Xist CHART-seq in WT and Smchd1$^{-/-}$ MEFs.

MEFs were cross-linked in PBS with 1% formaldehyde at room temperature for 10 min at 2 million cells/ml and then quenched with 0.125 M glycine at room temperature for 5 min. Cross-linked cells were washed three times with chilled PBST (1× PBS with 0.05% Tween-20) before being snap frozen in liquid nitrogen at 25 million cross-linked cells per pellet. In total, two frozen pellets were used for each experiment. Nuclei were prepared following a published protocol[28]. Because of the larger volume of the nuclei in MEFs, nuclei were gradually resuspended in chilled sonication buffer (50 mM HEPES, pH 7.5, 75 mM NaCl, 0.1 mM EGTA, 0.5% N-lauroylsarcosine, 0.1% sodium deoxycholate, 5 mM dithiothreitol, 10 U/ml SUPERaseIN) to reach approximately 270 µl final volume. Nuclei were then sonicated in two 130 µl batches within a Covaris microTUBE using Covaris E220e (140 W peak incident power, 10% duty factor, 200 cycles/burst). Sonicated chromatin was pooled and centrifuged at full speed for 20 min at 4 °C. The supernatant (~220 µl) was then pooled to reach ~440 µl final volume, which was then split into two CHART reactions (Xist capturing CHART and a negative control using sense probes)

following a protocol described previously (see Supplementary Data 3 for sequences of CHART probes)[28]. CHART-enriched DNA was eluted by 20 µl RNase H (5 U/µl; M0297S, NEB) in 200 µl elution buffer twice at 37 °C for 30 min. After validating CHART quality by quantitative PCR (qPCR) for the promoter of Xist and Gapdh (see Supplementary Data 3 for primer sequences), 100 µl input and CHART-enriched DNA were mixed with 30 µl TE buffer and further sheared to below 500 bp in a Covaris microTUBE for 2 min with Covaris E220e (140 W peak incident power, 10% duty factor, 200 cycles/burst). Sonicated DNA was purified and libraries were prepared by NEBNext® Ultra™ DNA Library Prep Kit for Illumina® (E7370S, NEB). Experiments were performed in two biological replicates. Input and CHART-seq libraries were sequenced on Illumina HiSeq, generating ~30 millions paired-end 50 nucleotide reads per sample.

### Analysis of Xist CHART-seq datasets.

Input and CHART-seq reads were processed as described in the section of ChIP-seq. After removing PCR duplicates, input-subtracted CHART profiles were generated using SPP, with smoothing using 1-kb windows recorded every 500 bp. Because RNA-seq analysis showed that WT and Smchd1$^{-/-}$ MEFs expressed Xist at the same level, and Xist RNA-FISH did not detect a dispersed Xist cloud (indicative of a chromatin-attachment defect) in Smchd1$^{-/-}$ cells, we extracted the X-chromosomal profiles from the Xist CHART profiles produced by SPP, and scaled the Xist coverage by equalizing the sum of Xist RNA coverage across the entire X chromosome in WT and Smchd1$^{-/-}$ MEFs using R. ΔXist profiles were then generated by subtracting scaled WT profiles from those of Smchd1$^{-/-}$ MEFs. Scaled Xist coverage profiles in WT and Smchd1$^{-/-}$ MEFs and ΔXist profiles were visualized using IGV, with scales indicated in each track. The density of Xist determined by CHART-seq at each X-linked genic and intergenic regions in WT or Smchd1$^{-/-}$ MEFs was computed by Homer on scaled CHART coverage profiles.

### Western blot.

Cells were washed once with PBS and lysed in chilled lysis buffer [10 mM Tris, pH 8.0, 1 M NaCl, 1% Triton, 0.5% sodium deoxycholate, 0.1% sodium dodecyl sulfate (SDS), 1 mM EDTA, 0.5 mM EGTA, 1× cOmplete EDTA-free PIC (11873580001, Sigma)]. Lysates were sonicated at 4 °C for 15 min (30 s ON; 30 s OFF) using Bioruptor®XL (Diagenode). Proteins were quantified using Bio-Rad Protein Assay (500-0006, Bio-Rad). Lysates (20–50 µg) were denatured in 1× Laemmli sample buffer at 95 °C for 5 min and resolved by SDS-polyacrylamide gel electrophoresis. Proteins were transferred from the gels to PVDF membranes (Immobilon Transfer Membrane; IPVH00010, EMD Millipore) in 1× TBE in a TE77XP Semi-dry Blotter (Hoefer) at 0.8 mA/cm$^2$ for 45 min. PVDF membranes were blocked with Blocking Buffer (5% milk and 0.1% Tween-20 in PBS) at room temperature for 1 h, incubated with antibodies against SMCHD1 (HPA039441, Sigma; 1:1000), RING1A (09-706, Millipore; 1:1000), RING1B (5694T, Cell Signaling; 1:1000), H2AK119ub (8240S, Cell Signaling; 1:2000), CTCF (2899S, Cell Signaling; 1:1000), or β-tubulin (T5201, Sigma, 1:2000) diluted in Blocking Buffer at 4 °C overnight. Membranes were then washed with PBST (PBS with 0.1% Tween-20) for 5 min three times, incubated with anti-rabbit IgG (H + L), HRP conjugate (W4011, Promega; 1:5000) or anti-mouse IgG (H + L), HRP conjugate (W4021, Promega; 1:10,000) diluted in Blocking Buffer at room temperature for 1 h, washed with PBST for 5 min three times, and developed using Western Lightning® Plus-ECL (NEL105001EA, Perkin-Elmer).

### PRC1 and HNRNPK KD.

One day prior to transfection, 0.6 million MEFs were plated on 10-cm plates. The next day, cells were transfected with either scramble siRNA (siGENOME Non-Targeting siRNA #1, D-001210-01, Dharmacon), siRNA targeting HNRNPK (L-048002-01, Dharmacon), or a mixture of siRNA targeting RING1A (D-062335-03, Dharmacon) and RING1B (D-042180-01, Dharmacon), using 30 µl Lipofectamine® RNAiMAX Transfection Reagent (13778075, Thermo Fisher Scientific). For HNRNPK KD, cells were transfected again after 24 h and harvested 3 days after the first transfection. For PRC1 KD, transfection was performed once a day for 4 days. Transfected cells were then used for in situ Hi-C, western blot, immuno-RNA-FISH, and quantitative reverse transcription-PCR (RT-qPCR).

### Screening for protein factors required for SMCHD1 recruitment.

To identify protein factors involved in recruiting SMCHD1 to the Xi, we used siRNA to deplete several factors reported in three previous studies that systemically identified Xist-interacting proteins[21,55,57]. After identifying HNRNPK, we tested if PRC1 and PRC2, two Polycomb complexes targeted to the Xi via HNRNPK[24,55,56], play a role in SMCHD1 recruitment. EY.T4 MEFs were transfected with either scramble control siRNA (D-001810-10-05, Dharmacon) or siRNA targeting HNRNPK (L-048002-01, Dharmacon), SPEN (M-062019-01-0005, Dharmacon), LBR (GS98386, QIAGEN), and RBM15 (GS229700, Qiagen), EED (L-049898-00-0005, Dharmacon), RING1A (D-062335-03, Dharmacon), and RING1B (D-042180-01, Dharmacon) using Lipofectamine® RNAiMAX Transfection Reagent (13778075, Thermo Fisher Scientific). As a positive control, we treated cells with siRNA targeting SMCHD1 (L-040501-01-0005, Dharmacon). At 72 h after transfection, immuno-RNA-FISH for SMCHD1 and Xist was performed to determine the effect of siRNA treatment on SMCHD1 localization. We verified KD efficiency by western blots using rabbit polyclonal HNRNPK antibody (11426-1-AP, Proteintech,

1:2000), rabbit polyclonal LBR antibody (12398-1-AP, Proteintech, 1:1000), rabbit polyclonal RBM15 antibody (ab70549, Abcam, 1:1000), rabbit polyclonal SMCHD1 antibody (HPA039441, Sigma, 1:1000), and rabbit monoclonal glyceraldehyde 3-phosphate dehydrogenase antibody (2118, Cell Signaling, 1:2000). KD of Ring1a, Ring1b, Eed, and Spen was verified by RT-qPCR.

**Quantitative reverse transcription-PCR**. RNA was extracted using TRIzol (15596018, Thermo Fisher Scientific) according to the manufacturer's instructions. Genomic DNA was removed from RNA using TURBO DNA-free™ Kit (AM1907, Thermo Fisher Scientific). After inactivating TURBO DNase by DNase Inactivation Reagent, DNase-treated RNA was reverse transcribed using SuperScript III Reverse Transcriptase (18080-085, Thermo Fisher Scientific) with 250 ng random primers (C118A, Promega) at 25 °C for 5 min, 50 °C for 1 h, and 85 °C for 15 min. Complementary DNA was 5–10-fold diluted with $H_2O$. qPCR was then performed using iTaq™ Universal SYBR® Green Supermix (1725125, Bio-Rad) in a CFX96 Real-Time PCR Detection System (Bio-Rad). To calculate the fraction of remaining transcripts in siRNA-treated cells, we normalized transcripts to β-actin, a housekeeping gene, and then set scramble siRNA-treated cells to 100%. See Supplementary Data 3 for primer sequences.

**Reporting summary**. Further information on research design is available in the Nature Research Reporting Summary linked to this article.

## Data availability
Raw and processed sequencing data have been deposited in the GEO accession GSE116413. The source data underlying Supplementary Figs. 1a, 11, 12a, 14b, and 14d are provided as a Source Data file. All other relevant data supporting the key findings of this study are available within the article and its Supplementary Information files or from the corresponding authors upon reasonable request.

## Code availability
The custom analysis pipelines for all genomic analyses are available upon request with no restrictions.

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

## Acknowledgements

We thank J.E. Froberg and A. Kriz for critical reading of the manuscript. We thank all lab members for intellectual support, U. Kim and the MGH Next Generation Sequencing Core, and M. Kuroda, S. Buratowski, and A. Gimelbrant for valuable advice during C-Y. W.'s thesis work. This work was supported by grants to J.T.L. from the National Institute of Health (RO1-GM090278), the Rett Syndrome Research Trust, and International Rett Syndrome Foundation, and Howard Hughes Medical Institute.

## Author contributions

C.-Y.W. and J.T.L. conceived the project, analyzed the data, and wrote the paper. C.-Y.W. performed all experiments and bioinformatics analyses. D.C. and H.S. contributed H2AK119ub ChIP-seq datasets for wild-type female MEFs. D.W. assisted with RT-qPCR, genotyping, and microscopy.

## Additional information

**Competing interests:** J.T.L. is a co-founder of and a scientific advisor for Translate Bio and Fulcrum Therapeutics. The other authors declare no competing interests.

