## [Peer Review File · Nature Communications]

REVIEWERS' COMMENTS:

Reviewer #3 (Remarks to the Author):

The authors have gone to great lengths to improve the manuscript, both by adding a considerable amount of new data and by rephrasing and reorganizing specific sections. Together, their efforts correctly address all the issues I previously raised.

I have three minor issues, related to the new data or the reorganization for the resubmission, but otherwise I support the publication of the manuscript.

Minor issues:

- Page 8: discussion of reactivation Class I genes vs other genes:

The authors have greatly improved the presentation of this data. I now agree that the data convincingly show that Class I genes are enriched for reactivation. In the discussion on page 8, where the authors correctly mention that not all Class I genes are reactivated, they should mention that a smaller fraction of Class II and III genes does get reactivated.

- Page 9 and Figure 4f: the intriguing observation that Xist distribution is more discrete in *Smchd1*^{-/-} cells.

With a challenging technique like CHART-seq, additional evidence may help to exclude the possibility of experimental variation, like varying signal-to-noise ratio between WT and mutant experiments. Here, visualization and quantitation of (supposedly absent) Xist binding to autosomes would be very helpful (like Fig. S6A for H3K27me3 ChIP).

- The order of figure panels in this version of the manuscript has at times become somewhat disorganized. To improve the structure of the figures, I propose the following reorganization:

>> Panel S1d is mentioned the first time on page 8. It will fit better when included in Fig. S6

>> Panel S3e is discussed after panel S4f. It will fit better in Figure S4.

>> Panels 5f, 5g and Panel S1e-g are all discussed in relationship to PRC1 (pages 10 and 11), which is the topic of Figures 6 and S7. They will fit better when grouped in those figures.

Reviewer #4 (Remarks to the Author):

This revision has included significant new experiments and addressed well the concerns from the previous reviewers. The findings that HNRNPK, PRC1 and SMCHD1 are required for coordinated folding of the inactive X chromosome in post-XCI cells are in agreement with several latest publications including one from the authors' lab (Sakakibara et al., 2018, Jansz et al., 2018, Gdula et al., 2019, Colognori et al., 2019) and provide additional new insights in maintaining of Xi chromatin structure and gene silencing, which is of great interest to the field. Only a few minor points need be clarified.

1. It would be helpful to describe the classification of Class I, II, III genes in the text or method section so that the readers don't need go to the original 2018 Cell paper looking for the definition.

2. The new experiments are poorly/incorrectly described or missed in the Methods. For example, "Xist CHART-sequencing in wild-type and *Smchd1*^{-/-} NPCs" should be in "wild-type and *Smchd1*^{-/-}

MEFs"? "in situ Hi-C on wild-type MEFs treated with scramble siRNAs or siRNAs targeting RING1A and RING1B or HNRNPK" is not mentioned. New in situ Hi-C in Xi Δ Xist fibroblasts is not mentioned.

3. Allele-specific ChIP-seq were done for H3K27me3 and H2AK119ub but only the whole X but not Xa or Xi profiles were shown (Fig. 3g and 6a, the legend states the profiles across the X chromosome). The authors should clarify this.

4. Incorrect legends for Fig. 6c-e and Fig. S1f-g.

Point-by-point response to reviewers' comments

Reviewer #3 (Remarks to the Author):

The authors have gone to great lengths to improve the manuscript, both by adding a considerable amount of new data and by rephrasing and reorganizing specific sections. Together, their efforts correctly address all the issues I previously raised.

I have three minor issues, related to the new data or the reorganization for the resubmission, but otherwise I support the publication of the manuscript.

We thank Rev3 for his/her constructive comments on our manuscript.

Minor issues:

- Page 8: discussion of reactivation Class I genes vs other genes:

The authors have greatly improved the presentation of this data. I now agree that the data convincingly show that Class I genes are enriched for reactivation. In the discussion on page 8, where the authors correctly mention that not all Class I genes are reactivated, they should mention that a smaller fraction of Class II and III genes does get reactivated.

Done.

- Page 9 and Figure 4f: the intriguing observation that Xist distribution is more discrete in *Smchd1*^{-/-} cells.

With a challenging technique like CHART-seq, additional evidence may help to exclude the possibility of experimental variation, like varying signal-to-noise ratio between WT and mutant experiments. Here, visualization and quantitation of (supposedly absent) Xist binding to autosomes would be very helpful (like Fig. S6A for H3K27me3 CHIP).

We thank Rev3 for this very insightful comment. We agree that CHART is a challenging technique and experimental variation must be taken into account when interpreting CHART data. We believed that the difference in CHART profiles between WT and *Smchd1*^{-/-} MEFs cannot be easily explained by experimental variation for two reasons.

- We performed Xist CHART on WT and *Smchd1*^{-/-} MEFs in two biological replicates, and observed the same trend (Supplementary Fig. 8b).
- For each CHART experiment, we also assessed capturing efficiency and specificity before library preparation and sequencing. This was done by performing qPCR using an aliquot of CHART-enriched DNA to measure the enrichment of the promoter of *Xist* (a positive control locus) and *Gapdh* (an autosomal locus as a negative control). In parallel, we also performed a control experiment using negative control probes that are antisense to the Xist capturing probes (Tsix probes). As shown in the figure below, we did not see varying signal-to-noise ratios between the CHART experiments done on WT and *Smchd1*^{-/-} MEFs. Indeed, in both biological replicates, CHART on both WT and *Smchd1*^{-/-} MEFs produced similarly high enrichment for the *Xist* promoter, which was >100-fold higher than the *Gapdh* promoter and ~10-fold higher than the *Xist* promoter in the control CHART experiment (labeled as "Tsix" in the figure below). Having validated the quality of our CHART experiments, we went on to prepare libraries from the remaining CHART-enriched DNA, which resulted in the CHART-seq data presented in the manuscript.

Together, these results lead us to believe that it is less likely that the differences in Xist distribution on the WT and *Smchd1*^{-/-} Xi reflect experimental variation of CHART. We have added Supplementary Figure 10 and relevant texts to address this point in the revised manuscript.

-The order of figure panels in this version of the manuscript has at times become somewhat disorganized. To improve the structure of the figures, I propose the following reorganization:

We have moved the figures accordingly as described in detail below.

>> Panel S1d is mentioned the first time on page 8. It will fit better when included in Fig. S6

Fig. S6 (Supplementary Fig. 8 in the revised manuscript) is already full. We have therefore moved Fig. S1d to Supplementary Fig. 9 in the revised manuscript.

>> Panel S3e is discussed after panel S4f. It will fit better in Figure S4.

Fig. S4 is already full. We have therefore moved Fig. S3e to Supplementary Fig. 5 in the revised manuscript.

>> Panels 5f, 5g and Panel S1e-g are all discussed in relationship to PRC1 (pages 10 and 11), which is the topic of Figures 6 and S7. They will fit better when grouped in those figures.

Fig. S7 (Supplementary Fig. 14 in the revised manuscript) is already full. We have therefore moved Fig. S1e-g to Supplementary Fig. 11 and 12 in the revised manuscript. We cannot move Fig. 5f and 5f, as Fig. 6 is already full. We believe that the current order will not compromise the readability of our manuscript.

Reviewer #4 (Remarks to the Author):

This revision has included significant new experiments and addressed well the concerns from the precious reviewers. The findings that HNRNPK, PRC1 and SMCHD1 are required for coordinated folding of the inactive X chromosome in post-XCI cells are in agreement with several latest publications including one from the authors' lab (Sakakibara et al., 2018, Jansz et al., 2018, Gdula et al., 2019, Cognori et al., 2019) and provide additional new insights in maintaining of Xi chromatin structure and gene silencing, which is of great interest to the field.

Only a few minor points need be clarified.

We thank Rev4 for his/her kind comments.

1. It would be helpful to describe the classification of Class I, II, III genes in the text or method section so that the readers don't need go to the original 2018 Cell paper looking for the definition.

We have included the definition of Class I, II, and III genes in the Methods section of the revised manuscript.

2. The new experiments are poorly/incorrectly described or missed in the Methods. For example, "Xist CHART-sequencing in wild-type and Smchd1-/- NPCs" should be in "wild-type and Smchd1-/- MEFs"? "in situ Hi-C on wild-type MEFs treated with scramble siRNAs or siRNAs targeting RING1A and RING1B or HNRNPK" is not mentioned. New in situ Hi-C in XiΔXist fibroblasts is not mentioned.

We have corrected the error and included the description of new experiments in the Methods section of the revised manuscript.

3. Allele-specific ChIP-seq were done for H3K27me3 and H2AK119ub but only the whole X but not Xa or Xi profiles were shown (Fig. 3g and 6a, the legend states the profiles across the X chromosome). The authors should clarify this.

The ChIP-seq profiles of H3K27me3 and H2AK119ub displayed in the manuscript are the ones generated from all reads ("comp" tracks), including allele-specific reads and reads that do not carry allelic information. As the majority of H3K27me3 and H3AK119ub ChIP-seq reads mapped to the X chromosome is contributed from the Xi, we believe that the comp tracks are representative of the Xi profile. Moreover, the comp tracks are not influenced by varying SNP densities along the X. By contrast, SNP densities could confound the interpretation of allele-specific tracks. Therefore, we decided to display the comp tracks, but not the allele-specific tracks, in our manuscript. We have clarified this point in the legend of Fig. 3 and 6, as well as in the Methods section, of the revised manuscript. Both the comp and allele-specific tracks, which can be directly loaded onto the genome browser, have been deposited to GEO.

4. Incorrect legends for Fig. 6c-e and Fig. S1f-g.

We have provided correct legends for Fig. 6c-e and Fig. S1f-g (now Supplementary Fig. 12a,b) in the revised manuscript.